# On Differentially Private Sampling from Gaussian and Product Distributions

**Badih Ghazi**
Google Research
Mountain View, CA, US
badihghazi@gmail.com

**Xiao Hu**[*]
University of Waterloo
Waterloo, Canada
xiaohu@uwaterloo.ca

**Ravi Kumar**
Google Research
Mountain View, CA, US
ravi.k53@gmail.com

**Pasin Manurangsi**
Google Research
Bangkok, Thailand
pasin@google.com

## Abstract

Given a dataset of $n$ i.i.d. samples from an unknown distribution $P$, we consider the problem of generating a sample from a distribution that is close to $P$ in total variation distance, under the constraint of differential privacy (DP). We study the problem when $P$ is a multi-dimensional Gaussian distribution, under different assumptions on the information available to the DP mechanism: known covariance, unknown bounded covariance, and unknown unbounded covariance. We present new DP sampling algorithms, and show that they achieve near-optimal sample complexity in the first two settings. Moreover, when $P$ is a product distribution on the binary hypercube, we obtain a pure-DP algorithm whereas only an approximate-DP algorithm (with slightly worse sample complexity) was previously known.

## 1  Introduction

Differential privacy (DP) [18, 16] is a strong and rigorous notion of privacy that has been increasingly studied and deployed as protection against the leakage of personal data used to train ML models.

A basic setting widely studied in ML is *distribution learning*, where given samples drawn i.i.d. from an unknown distribution $P$, we seek to output a distribution $Q$ that is as close to $P$ as possible (formal definitions are given in Section 1.1). Recent works [27, 2, 29, 4] have studied DP *distribution learning*, whereby the output distribution is guaranteed to stay roughly the same when a single input sample is changed. A closely related setting is DP mean and covariance estimation studied by [38, 21, 36, 30, 7, 8, 11, 15, 28, 25, 9, 26]. Motivated by the fact that tasks often require much fewer samples than full-fledged learning, the very recent work of [35] studied the task of DP *distribution sampling*, where the goal is to generate a sample from a distribution $Q$ that is as close as possible to the distribution $P$ from which the input samples are drawn i.i.d. (In the non-private setting, variants of the sampling task have also been studied, e.g., in the work on sample amplification by [5].)

In this work, we study DP distribution sampling, and the quantitative gap with respect to the *a priori* more challenging task of DP distribution learning. For the case of multi-dimensional Gaussians, we consider three natural settings: known covariance, unknown bounded covariance, and unknown unbounded covariance; for the first two, we obtain near-tight bounds. This answers an open question of [35]. Moreover, we obtain near-tight bounds on sampling in the case of product distributions on $\{0,1\}^d$ with pure-DP, also improving upon previous work.

---

[*]This work was done while the author was visiting Google Research.

37th Conference on Neural Information Processing Systems (NeurIPS 2023).

**Motivation.** There are many natural settings where a sample from the underlying distribution would be sufficient (as an alternative to the possibly more expensive task of learning the distribution).

For example, if one is implementing an algorithm that would run on sensitive user data, then, as part of the usual software development cycle, unit tests [24] are important. Writing unit tests on real data can violate users' privacy (e.g., if the unit test code is public). On the other hand, (privately) learning the distribution of the user data could be a significant overkill for this application. A private sample of the underlying distribution would provide a sufficiently representative input for the unit testing algorithm, and this input could be revealed publicly without compromising the privacy of the users.

A second motivation arises in distributed settings. Consider the situation where algorithms are being developed on medical data provided by multiple hospitals. The algorithm designer would like a synthetic dataset that would work well for the algorithm development process. Such a dataset could be obtained if each hospital provides a few (private) samples from its underlying distribution.

Finally, we describe another distributed setting motivation where a central curator wishes to build a synthetic dataset from multiple users, each contributing multiple items, under the constraint of item-level DP (as opposed to user-level DP). A simple approach for generating such a synthetic dataset is to let each user send to the curator a private sample (satisfying item-level DP).

## 1.1 Formulation

Let $\mathcal{D}$ be a class of distributions on some domain $\mathcal{X}$. We consider a setting where there is an unknown distribution $D \in \mathcal{D}$ and an algorithm has sample access to $D$. There are two natural problems that can be posed in this setting. In the *sampling problem*, the goal is to design an algorithm that uses samples from $D$ (which is unknown) and outputs an element in $\mathcal{X}$. We say that an algorithm $\mathcal{A}$ is an $\alpha$-*accurate sampler for* $\mathcal{D}$ iff $d_{\mathrm{tv}}(D, Q_{\mathcal{A},D}) \leq \alpha$ for all $D \in \mathcal{D}$, where $Q_{\mathcal{A},D}$ denotes the distribution of $\mathcal{A}(\mathbf{X})$ where $\mathbf{X} \sim D^n$ and $d_{\mathrm{tv}}(\cdot, \cdot)$ denotes the total variation distance. Here $n$ is said to be the sample complexity of the sampler. In the *learning problem*, the goal is to design an algorithm that outputs a distribution $D' \in \mathcal{D}$. We say that an algorithm $\mathcal{A}$ is an $(\alpha, \beta)$-*accurate learner for* $\mathcal{D}$ if $\Pr_{\mathbf{X} \sim D^n, D' \sim \mathcal{A}(\mathbf{X})}[d_{\mathrm{tv}}(D', D) \leq \alpha] \geq 1 - \beta$ for all $D \in \mathcal{D}$; as before, $n$ is the sample complexity of the learner.

The DP version of the learning problem is well studied, e.g., [27, 4]; in this work we study the DP version of the sampling problem. First, we recall the definition of DP. We consider the *substitution* notion, i.e., two datasets are *neighbors* iff they have the same number of samples and we can transform one to the other by changing a single sample.

**Definition 1.1** (Differential Privacy [18, 16]). An algorithm $M : \mathcal{Y} \to \mathcal{O}$ is said to be $(\epsilon, \delta)$-*differentially private* $((\epsilon, \delta)$-*DP)* for $\epsilon > 0, \delta \geq 0$ iff, for every $S \subseteq \mathcal{O}$ and every neighboring datasets $Y, Y' \in \mathcal{Y}$, we have $\Pr[\mathcal{M}(Y) \in S] \leq e^{\epsilon} \cdot \Pr[\mathcal{M}(Y') \in S] + \delta$.

We abbreviate $(\epsilon, 0)$-DP by $\epsilon$-DP, aka, *pure-DP*; the $\delta \neq 0$ case is *approximate-DP*. We assume that the privacy parameters satisfy $\epsilon \leq 1, \delta \leq 1/2$, and the accuracy parameter satisfies $0 < \alpha \leq 1/2$.

## 1.2 Our Results for Gaussian Distributions in $\mathbb{R}^d$

We focus on the $d$-dimensional Gaussian distribution, i.e., $D = \mathsf{N}(\mu, \Sigma)$ with $\mu \in \mathbb{R}^d, \Sigma \in \mathbb{R}^{d \times d}$. Recall that $\mathsf{N}(\mu, \Sigma)$ is supported on $\mathbb{R}^d$ with $f_{\mathsf{N}(\mu,\Sigma)}(x) \propto \exp\left(-\frac{1}{2}(x - \mu)^T \Sigma^{-1} (x - \mu)\right)$. We study three cases that have been considered in the literature:

- *Known Covariance:* $\Sigma$ is known, but $\mu$ is not. More formally, the class of distributions is $\mathcal{D}_{\Sigma}^{\mathsf{N}} := \{\mathsf{N}(\mu, \Sigma) \mid \mu \in \mathbb{R}^d\}$.
- *Unknown Bounded Covariance:* Both $\mu, \Sigma$ are unknown but with a promise that $I \preceq \Sigma \preceq \kappa \cdot I$ for a known constant $\kappa > 0$, i.e., the class of distributions is $\mathcal{D}_{\kappa}^{\mathsf{N}} := \{\mathsf{N}(\mu, \Sigma) \mid \mu \in \mathbb{R}^d, \Sigma \in \mathbb{R}^{d \times d} \text{ s.t. } I \preceq \Sigma \preceq \kappa \cdot I\}$.[2]
- *Unknown Unbounded Covariance:* Both $\mu, \Sigma$ are unknown, i.e., the class of distributions is $\mathcal{D}^{\mathsf{N}} := \{\mathsf{N}(\mu, \Sigma) \mid \mu \in \mathbb{R}^d, \Sigma \in \mathbb{R}^{d \times d}\}$.

A summary of our DP sampling results along with a comparison to known DP learning results is in Table 1. Before we describe our results in more detail, we highlight the following. (i) In all cases,

---

[2]We write $A \preceq B$ to denote that $B - A$ is positive semi-definite, for matrices $A$ and $B$.

the dependence of our algorithms on the accuracy parameter $\alpha$ is only polylogarithmic, whereas for DP learning algorithms, this dependence is polynomial. (ii) In the case of known covariance and unbounded known covariance, we obtain improvements over DP learning in terms of the dependence on the dimension $d$. (iii) All of our algorithms run in polynomial time, although we do not explicitly state the running time in the formal statements.

**Known Covariance.** While any DP learning algorithm requires $\tilde{\Theta}\left(\frac{d}{\alpha^2} + \frac{d}{\alpha\epsilon}\right)$ samples in this setting [27], we show that, surprisingly, only $\tilde{O}\left(\sqrt{d}/\epsilon\right)$ samples suffice for DP sampling.

**Theorem 1.2.** *There is an $\alpha$-accurate $(\epsilon, \delta)$-DP sampler for Gaussian distributions with known covariance with sample complexity $O\left(\frac{\sqrt{d}}{\epsilon} \cdot \text{polylog}\left(\frac{d}{\delta\epsilon\alpha}\right)\right).$*

We also show that the $\sqrt{d}$ dependence is necessary, i.e., that our algorithm's sample complexity is tight up to polylogarithmic factors.

**Theorem 1.3.** *Any 0.1-accurate $(\epsilon, \delta)$-DP sampler for Gaussian distributions with known covariance must have sample complexity $\Omega(\sqrt{d}/\epsilon)$.*

**Unknown Bounded Covariance.** In this setting, we obtain an algorithm with sample complexity of $\tilde{O}_\kappa\left(\frac{d}{\epsilon}\right)$; in contrast, the best known DP learning algorithm requires $\tilde{\Theta}_\kappa\left(\frac{d^2}{\alpha^2} + \frac{d^2}{\alpha\epsilon}\right)$ samples [27, 29].

**Theorem 1.4.** *There is an $\alpha$-accurate $(\epsilon, \delta)$-DP sampler for Gaussian distributions with unknown covariance, under the assumption $I \preceq \Sigma \preceq \kappa \cdot I$, with sample complexity $O\left(\frac{d}{\epsilon} \cdot \kappa^2 \cdot \text{polylog}\left(\frac{d}{\delta\epsilon\alpha}\right)\right).$*

Similar to before, we can show that the sample complexity dependence on $d, \epsilon$ is near-optimal:

**Theorem 1.5.** *Let $\alpha > 0$ be a sufficiently small constant, and $\epsilon, \delta$ be such that $\delta \leq O\left(\frac{1}{nd^2}\right)$. Then, any $\alpha$-accurate $(\epsilon, \delta)$-DP sampler for Gaussian distributions with unknown covariance, under the assumption $I \preceq \Sigma \preceq 2I$, must have sample complexity $n = \Omega\left(\frac{d}{\epsilon\sqrt{\log d}}\right).$*

**Unknown Unbounded Covariance.** In this setting, the best known DP learning algorithm uses $\tilde{\Theta}\left(\frac{d^2}{\alpha^2} + \frac{d^2}{\alpha\epsilon}\right)$ samples [4, 29, 28]. For DP sampling, we show that we can reduce the dependence on $\alpha$ to polylogarithmic and the dependence on $d$ to $d^{1.5}$.

**Theorem 1.6.** *There exists an $\alpha$-accurate $(\epsilon, \delta)$-DP sampler for Gaussian distributions (without any assumption) with sample complexity $O\left(\frac{d^{1.5}}{\epsilon} \text{polylog}\left(\frac{d}{\alpha\epsilon\delta}\right)\right).$*

|  | Known Covariance | Bounded Covariance | Unbounded Covariance |
|---|---|---|---|
| Non-Private Learning (Folklore) | $\Theta\left(\frac{d}{\alpha^2}\right)$ | $\Theta\left(\frac{d^2}{\alpha^2}\right)$ | $\Theta\left(\frac{d^2}{\alpha^2}\right)$ |
| $(\epsilon, \delta)$-DP Learning | $\tilde{\Theta}\left(\frac{d}{\alpha^2} + \frac{d}{\alpha\epsilon}\right)$ [27] | $\tilde{\Theta}\left(\frac{d^2}{\alpha^2} + \frac{d^2}{\alpha\epsilon}\right)$ [27, 28] | $\tilde{\Theta}\left(\frac{d^2}{\alpha^2} + \frac{d^2}{\alpha\epsilon}\right)$ [4, 28] |
| $(\epsilon, \delta)$-DP Sampling (Our results) | $\tilde{\Theta}\left(\frac{\sqrt{d}}{\epsilon}\right)$ Theorems 1.2,1.3 | $\tilde{\Theta}\left(\frac{d}{\epsilon}\right)$ Theorems 1.4, 1.5 | $\tilde{O}\left(\frac{d^{1.5}}{\epsilon}\right)$ Theorem 1.6 |

Table 1: Sample complexity of private learning and sampling for Gaussian distributions. Here, $\tilde{O}, \tilde{\Theta}$ hide factors that are polylogarithmic in $d, 1/\epsilon, 1/\delta, 1/\alpha$ (and $1/\beta$ in the case of learning).

## 1.3 Our Results for Product Distributions on $\{0, 1\}^d$

For $p \in [0, 1]$, let $\text{Ber}(p)$ be the *Bernoulli* distribution supported on $\{0, 1\}$ with probability mass function $f_{\text{Ber}(p)}(0) = 1 - p$ and $f_{\text{Ber}(p)}(1) = p$. We consider product distributions $\text{Ber}(p_1) \otimes \cdots \otimes \text{Ber}(p_d)$ where $p_1, \ldots, p_d \in [0, 1]$ are unknown. (In other words, the class of distributions is $\mathcal{D}^{\text{prod}} := \{\text{Ber}(p_1) \otimes \cdots \otimes \text{Ber}(p_d) \mid p_1, \ldots, p_d \in [0, 1]\}$.)

We give a pure-DP sampler with sample complexity $\tilde{O}\left(\frac{d}{\alpha\epsilon}\right)$. Previously, only approximate-DP sampler with similar sample complexity was known from [35], which also provided a matching lower bound. In comparison, DP learning uses $\tilde{\Theta}\left(\frac{d}{\alpha^2} + \frac{d}{\alpha\epsilon}\right)$ samples [27, 10].

| Non-Private Learning | $\Theta\left(\frac{d}{\alpha^2}\right)$ |
|---|---|
| $\epsilon$-DP Learning | $\tilde{\Theta}\left(\frac{d}{\alpha^2} + \frac{d}{\alpha\epsilon}\right)$[29, 27] |
| $(\epsilon, \delta)$-DP Sampling | $\tilde{\Theta}\left(\frac{d}{\alpha\epsilon}\right)$ [35] |
| $\epsilon$-DP Sampling (Our result) | $\tilde{\Theta}\left(\frac{d}{\alpha\epsilon}\right)$ Theorem 1.7 |

Table 2: Sample complexity for private learning and sampling for product distributions on $\{0,1\}^d$. Here, $\tilde{\Theta}$ hides factors that are polylogarithmic in $d, 1/\alpha$ (and $1/\beta$ in the case of learning).

**Theorem 1.7.** *There exists an $\alpha$-accurate $\epsilon$-DP sampler for product distributions on $\{0,1\}^d$ with sample complexity $O\left(\frac{d\log\left(\frac{d}{\alpha}\right)}{\alpha\epsilon} + \frac{d\log^2\left(\frac{d}{\alpha}\right)}{\epsilon}\right)$.*

We also note that our result above improves upon even the approximate-DP sampler in [35] by logarithmic factors. Specifically, for $\alpha \leq 1/\log d$, our sample complexity is $O\left(\frac{d\log(d/\alpha)}{\alpha\epsilon}\right)$ whereas theirs is $O\left(\frac{d\sqrt{\log(1/\delta)}}{\alpha\epsilon}\left(\log^{9/4}d + \log^{5/4}(1/\alpha)\right)\right)$.

## 2 Technical Overview

### 2.1 Gaussian Distributions: Algorithms

**Known Covariance.** When $\Sigma$ is known, we may assume w.l.o.g. that $\Sigma = I$; otherwise, we can transform each sample $X$ into $\Sigma^{-1/2}X$. We start by using known algorithms [23, 40, 33] to find a "rough" estimate for the mean. In particular, we find an estimate $\hat{\mu}$ such that $\|\hat{\mu}-\mu\|_2 \leq R = \tilde{O}(\sqrt{d}/\epsilon)$ using $n = \tilde{O}(\sqrt{d}/\epsilon)$ samples. By appropriately shifting the subsequent samples, this is equivalent to assuming that $\|\mu\|_2 \leq R$. We then focus on designing a DP sampler for this bounded mean case. It turns out, surprisingly, that the Gaussian mechanism suffices here. Specifically, for a parameter $B > 0$ (chosen later), we truncate each sample so that its $\ell_2$-norm is at most $B$. We then output their average with a (spherical) Gaussian noise $\mathsf{N}(0, \sigma^2 I)$ added. The description is given in Algorithm 1.

The analysis of the Gaussian mechanism [e.g., 20, Appendix A] shows that the algorithm is $(\epsilon, \delta)$-DP as long as we pick $\sigma \geq \tilde{O}\left(\frac{B}{n\epsilon}\right)$.

As for the accuracy, observe that if there were no truncation, then the output is exactly distributed as $\mathsf{N}(\mu, (\sigma^2 + 1/n)I)$, which is precisely $\mathsf{N}(\mu, I)$ if we set $\sigma^2 = (n-1)/n$. Therefore, by setting $B = R + O(\sqrt{d + \log(1/\alpha)})$ so that the truncation does *not* occur with probability $1 - \alpha$, we

---

**Algorithm 1** SPHERICALGAUSSIANSAMPLER
___
**Parameters:** $B, \sigma > 0$, and $n \in \mathbb{N}$.
Sample $X_1, \ldots, X_n \sim D$
**for** $i = 1, \ldots, n$ **do**
$\quad\mid\quad X_i^{\text{trunc}} = \text{trunc}_B^2(X_i) \qquad\qquad \triangleright$ see (1)
Sample $Z \sim \mathsf{N}(0, \sigma^2 I)$
**return** $Z + \frac{1}{n}\sum_{i\in[n]} X_i^{\text{trunc}}$

---

ensure that the sampler is $\alpha$-accurate. The constraint that $\sigma \geq \tilde{O}\left(\frac{B}{n\epsilon}\right)$ from privacy implies that we need $n \geq \tilde{O}(\sqrt{d}/\epsilon)$ and hence yielding the sample complexity in Theorem 1.2.

**Unknown Bounded Covariance.** Recall that in this setting we know that $I \preceq \Sigma \preceq \kappa \cdot I$. While it might be tempting to use the above Gaussian mechanism for this setting as well, it turns out that this approach results in sample complexity that depends *polynomially* on $1/\alpha$.[3]

To circumvent this, we first consider the case where $\mu = 0$ (i.e., "centered" Gaussians). In this case, our algorithm originates from the following attempt: output $\sum_{i\in[n]} a[i] \cdot X_i$, where $(a[1], \ldots, a[n]) \sim \mathsf{Uni}_n^{\mathbb{S}}$, the uniform distribution over points on the unit sphere in $\mathbb{R}^n$. It follows from the 2-stability of the Gaussian distribution [41] that, when $X_1, \ldots, X_n \sim \mathsf{N}(0, \Sigma)$, this results[4] in an output that is distributed exactly as $\mathsf{N}(0, \Sigma)$.

Unfortunately, this algorithm is not DP: if $X_1 = \cdots = X_{n-1} = 0$, then the output will reveal the direction of $X_n$ in the clear. To remedy this, we build on the intuition that, if $X_1, \ldots, X_n$ "sufficiently span all directions", then there should be "enough noise" to make this algorithm DP. In particular, using the bounded covariance property, we can show that if all the eigenvalues of $\sum_{i\in[n]} X_i X_i^T$

---

[3]See Appendix C for a proof sketch of the sample complexity from such an approach.
[4]Note that this holds even for any fixed unit vector $a \in \mathbb{R}^d$.

are sufficiently large (and each $X_i$ is truncated appropriately), then this algorithm is indeed "DP". This is perhaps the most technically challenging part of our work, as the noise is data-dependent and therefore poses significant hurdles in the privacy analysis (Section 4.2). Note that this is also the reason we need $n \geq \Omega(d)$, as otherwise $X_1, \ldots, X_n$ cannot "sufficiently span all directions" in $\mathbb{R}^d$.

With the above, the last ingredient is a testing step (in the "propose-test-release" paradigm of [17]) that checks this eigenvalue condition. When this condition fails, we return $\perp$; otherwise $\sum_{i \in [n]} a[i] \cdot X_i$.

To handle the case where $\mu \neq 0$, we take an output to be the sum of the average of $n_1$ samples and $\sqrt{1 - \frac{1}{n_1}} \cdot (\sum_{i \in [n_2]} a[i] \cdot U_i)$, where each $U_i$ is the difference between two fresh independent samples divided by $\sqrt{2}$ and $(a[1], \ldots, a[n_2]) \sim \mathsf{Uni}_{n_2}^{\mathbb{S}}$. Notice here that each $U_i \sim \mathsf{N}(0, \Sigma)$, while the average over $n_1$ samples is $\sim \mathsf{N}(\mu, \frac{1}{n_1} \cdot \Sigma)$; thus, the sum is $\sim \mathsf{N}(\mu, \Sigma)$ as desired. The full description is presented in Algorithm 2; the parameter setting and analysis can be found in Appendix B.4.

**Unknown Unbounded Covariance.** We proceed by reducing this case to the previous setting. We do so by first applying the known DP "preconditioner" algorithm for Gaussians (with unknown unbounded covariance) from the work of [9] in order to obtain rough estimates $\hat{\mu}, \hat{\Sigma}$ of $\mu, \Sigma$ respectively. This allows us to transform any subsequent sample $X$ into $\hat{\Sigma}^{-1/2}(X - \hat{\mu})$. This reduces us back to DP sampling for $\mathsf{N}(\hat{\Sigma}^{-1/2}(\mu - \hat{\mu}), \hat{\Sigma}^{1/2}\Sigma^{-1}\hat{\Sigma}^{1/2})$. The guarantee of the DP preconditioner ensures that this Gaussian actually has bounded

---

**Algorithm 2** BOUNDEDCOVGAUSSIANSAMPLER

**Parameters:** $B, \Delta > 0$, and $n_1, n_2 \in \mathbb{N}$.
Sample $X_1, \ldots, X_{n_1}, X_{n_1+1}, \ldots, X_{n_1+2n_2} \sim D$
**for** $i = 1, \ldots, n_1 + 2n_2$ **do**
$\quad | \quad X_i^{\mathsf{trunc}} = \mathsf{trunc}_B^2(X_i)$ $\qquad\qquad$ ▷ see (1)
**for** $j = 1, \ldots, n_2$ **do**
$\quad | \quad U_i = \frac{1}{\sqrt{2}}(X_{n_1+2i-1}^{\mathsf{trunc}} - X_{n_1+2i}^{\mathsf{trunc}})$
Sample $r \sim \mathsf{STLap}\left(\frac{\epsilon}{2}, \frac{\delta}{2}, \Delta\right)$ $\qquad$ ▷ see Lemma 3.1
**if** $\lambda_{\min}\left(\sum_{i \in [n_2]} U_i U_i^T\right) + r \geq 0.75n_2$ **then**
$\quad | \quad$ **return** $\perp$
Sample $a \sim \mathsf{Uni}_{n_2}^{\mathbb{S}}$
**return** $\frac{1}{n_1}\left(\sum_{i \in [n_1]} X_i^{\mathsf{trunc}}\right) + \sqrt{1 - \frac{1}{n_1}} \cdot (\sum_{i \in [n_2]} a[i] \cdot U_i)$

---

covariance. Therefore, we can apply our previous algorithm. Note that a significant part of the sample complexity is due to the DP preconditioner of the Gaussian; it turns out that the sample complexity of this task is only $\tilde{O}_{\epsilon,\alpha}(d^{3/2})$ (compared to $\tilde{\Theta}_{\epsilon,\alpha}(d^2)$ for learning). Furthermore, since we only need the preconditioner to be a rough estimate, we can set the accuracy parameter for the learning to be $\Theta(1)$ and thus avoid the polynomial dependence on $1/\alpha$. (Note that private preconditioner is a standard ingredient in the recipe for DP learning [e.g., 27].)

## 2.2 Gaussian Distributions: Lower Bounds

**Known Covariance.** Our lower bound in this setting builds on the following insight: if we take a constant number of samples from $\mathsf{N}(\mu, I)$ (using the DP sampler) and use them to estimate $\mu$, then we incur an expected $\ell_2^2$-error that is $O(d)$. It turns out that known lower bounds for DP mean estimation of Gaussians with known covariance [27] hold for this setting and give a lower bound of $\Omega\left(\frac{d}{\gamma\epsilon}\right)$, where $\gamma^2$ is the $\ell_2^2$-error. Plugging in $\gamma = \sqrt{d}$ in our setting gives the desired lower bound of $\Omega(\sqrt{d}/\epsilon)$. In the actual proof, one complication stems from the fact that our DP sampler does not output a sample exactly from $\mathsf{N}(\mu, I)$. Nonetheless, we can quantify this in terms of the accuracy $\alpha$.

**Unknown Bounded Covariance.** In this setting, we reduce from a lower bound on DP covariance estimation [28]—rather than DP mean estimation earlier—of centered Gaussians. The challenge is that $\Theta(1)$ samples from $\mathsf{N}(0, \Sigma)$ do *not* provide a sufficiently high accuracy estimate for $\Sigma$ so that we can apply the known lower bound[5]. Therefore, the above approach does not work directly.

To overcome this, we will have to use many samples to estimate the covariance. Recall that the sample complexity lower bound of $\Omega(d^2/\epsilon)$ for covariance estimation requires the accuracy in the Frobenius distance (or the Mahalanobis distance) to be constant [28]. Due to this accuracy requirement, we need to use our DP sampler to generate $L = \Omega(d^2)$ samples $Y_1, \ldots, Y_L$ to achieve such an accuracy. We can draw a fresh batch $X_1^i, \ldots, X_n^i$ of samples from the underlying distribution to generate each

---

[5]This high accuracy requirement is inherent in the known DP covariance estimation lower bound; see [28, Remark 4.4] for more details.

$Y_i$. However, since $L = \Omega(d^2)$, this would use $Ln$ samples in total. The covariance estimation lower bound would then yield $Ln \geq \Omega(d^2/\epsilon)$, implying $n \geq \Omega(1/\epsilon)$—even weaker than the mean estimation lower bound!

Fortunately, it turns out this can be overcome by using advanced composition of DP [19]. In particular, we may run our sampler $L$ times on the *same* $n$ samples to produce $Y_1, \ldots, Y_L$. In this case, the final covariance estimation algorithm has privacy loss parameter $\tilde{O}(\sqrt{L} \cdot \epsilon) = \tilde{O}(d \cdot \epsilon)$ due to advanced composition (Theorem A.5). Therefore, we get a lower bound of $\tilde{\Omega}(d^2/(d \cdot \epsilon)) = \tilde{\Omega}(d/\epsilon)$ as desired.

While the above overview seems intuitively plausible, there are certain difficulties that we need to overcome. First, the samples $Y_i$ that our algorithm produces are not exactly drawn from $\mathsf{N}(0, \Sigma)$; to fix this, we run an agnostic learner for Gaussians [e.g., 3] to recover the estimate of $\Sigma$. Second, since we run our DP sampler on the same $n$ samples, the produced $Y_1, \ldots, Y_L$ are not independent. We fix this by first drawing a larger number $N$ of samples, and then produce each $Y_i$ using $n$-out-of-$N$ random samples; this reduces the correlation across the $Y_i$'s. Furthermore, using amplification-by-sampling of DP [6] gives us the desired privacy-vs-sample complexity lower bound guarantee.

### 2.3 Product Distributions: Algorithm

Our sampler follows the framework of [35], which is built upon the preconditioning procedure proposed by [27] for DP learning.

We start with a private preconditioner, which obtains a crude estimate of each $p_j$ (up to a constant multiplicative factor). This is similar to that of [27], except that we use Laplace noise instead of Gaussian noise; this ensures that the resulting algorithm is pure-DP[6]. By suitably partitioning $[0, 1]$ into geometrically decreasing buckets in terms of $d/\alpha$, the goal then is to estimate $p_j$ by placing it in one of these buckets. This can be done by an appropriate thresholding and the Laplace mechanism.

The next step is to obtain a refined estimate of the $p_i$'s using a fresh batch of samples. The algorithm then returns a sample randomly drawn from the product distribution given by these estimates. The earlier crude estimates are helpful in truncating and clipping the samples to make the produced sample DP, without adding any noise to the refined estimate.

## 3 Preliminaries

For convenience, we use the notation $\mathrm{trunc}_B^p(X)$ for "truncation" for all $X \in \mathbb{R}^d, B > 0, p \geq 1$:

$$\mathrm{trunc}_B^p(X) := \begin{cases} X & \text{if } \|X\|_p \leq B, \\ X \cdot B/\|X\|_p & \text{if } \|X\|_p > B. \end{cases} \tag{1}$$

Let $[k]$ denote $\{1, \ldots, k\}$. For any $X \in \mathbb{R}^d$, we use $X[j]$ to denote the value of its $j$th coordinate.

### 3.1 Distributions and Tail Bounds

For a discrete distribution $D$, we use $f_D$ to denote its probability mass function (PMF); for a continuous distribution $D$, we use $f_D$ to denote its probability density function (PDF). Let $\mathrm{supp}(D)$ denote the support of $D$. We let $Z \sim D$ denote that the random variable $Z$ is distributed according to $D$; throughout, we may write the random variable in place of the distribution and vice versa when convenient. Finally, when $D_1, \ldots, D_d$ are distributions, we use $D_1 \otimes \cdots \otimes D_d$ as the product distribution, i.e., the distribution of $(Z_1, \ldots, Z_d)$ where $Z_1 \sim D_1, \ldots, Z_d \sim D_d$ are independent.

For a distribution $D$ on $\mathbb{R}^d$ and $v \in \mathbb{R}^d$, we write $D + v$ as a shorthand for the distribution of $X + v$ where $X \sim D$. Furthermore, for a distribution $D$ and a (possibly randomized) function $h$, we write $h(D)$ to denote the distribution of $h(X)$ where $X \sim D$.

We will list a few distributions that will be useful for us.

---

[6]We remark that a similar analysis has also been done by [37], who uses such a pure-DP preconditioner to give pure-DP algorithms for *learning* product distributions.

(i) *Shifted Truncated Discrete Laplace Distribution:* For $\Delta > 0$, let $s(\epsilon, \delta) = \lceil \Delta(1 + \log(1/\delta)/\epsilon) \rceil$. We define $\mathsf{STLap}(\epsilon, \delta, \Delta)$ to be the discrete distribution supported on $[-2s(\epsilon, \delta), 0]$ such that $f_{\mathsf{STLap}(\epsilon, \delta, \Delta)}(x) \propto \exp\left(-\epsilon \left| x + s(\epsilon, \delta) \right|\right)$.

It is known that adding $\mathsf{STLap}$ noise to a low-sensitivity function results in a DP estimate [e.g., 22].

**Lemma 3.1.** *If $g$ is a function with sensitivity $\leq \Delta$, then the algorithm that outputs $g(\mathbf{X}) + \mathsf{STLap}(\epsilon, \delta, \Delta)$ is $(\epsilon, \delta)$-DP.*

(ii) *Beta Distribution:* For $\alpha, \beta > 0$, $\mathsf{Beta}(\alpha, \beta)$ has the PDF $f_{\mathsf{Beta}(\alpha, \beta)}(x) \propto x^{\alpha - 1}(1 - x)^{\beta - 1}$.

(iii) *Uniform Distribution over Unit Sphere:* For $d \in \mathbb{N}$, let $\mathsf{Uni}_d^{\mathbb{S}}$ denote the distribution of a random unit vector in $\mathbb{R}^d$.

(iv) *Projection of Uniform Distribution over Unit Sphere:* For any $d \in \mathbb{N}$ and $i \in [d]$ and $z \in \mathbb{R}^d$, let $\Pi_{\leq i}(z)$ denote $(z_1, \ldots, z_i)$. Then, let $\mathsf{Uni}_{d,i}^{\mathbb{S}}$ denote the distribution of $\Pi_{\leq i}(Z)$ where $Z \sim \mathsf{Uni}_d^{\mathbb{S}}$. This distribution has the PDF (see, e.g., [34, Theorem 2]) given below.

$$f_{\mathsf{Uni}_{d,i}^{\mathbb{S}}}(z) \propto \begin{cases} (1 - \|z\|^2)^{\frac{d-i}{2} - 1} & \text{if } \|z\|^2 < 1 \\ 0 & \text{otherwise.} \end{cases} \tag{2}$$

We need a tail bound on the $\ell_2$-norm of a Gaussian-distributed vector:

**Lemma 3.2** ([42, Theorem 6.2]). *There exists a constant $c \geq 1$ such that, for any $\mu \in \mathbb{R}^d$, $\Sigma \in \mathbb{R}^{d \times d}$ where $\Sigma \preceq \kappa \cdot I$ and any $\beta \in (0, \frac{1}{2})$, $\Pr_{X \sim \mathsf{N}(\mu, \Sigma)}\left[ \|X - \mu\|_2 > c\sqrt{\kappa}\left(\sqrt{d} + \sqrt{\log(1/\beta)}\right) \right] \leq \beta$.*

We also need a tail bound for the beta distribution:

**Lemma 3.3** ([43], Theorem 8). *There exists a constant $c \in (0, 1)$ such that, for any $0 < \alpha < \beta$ and any $x \geq 0$, we have $\Pr_{Z \sim \mathsf{Beta}(\alpha, \beta)}\left[ Z \geq \frac{\alpha}{\alpha + \beta} + x \right] \leq 2e^{-c \cdot \min\left\{ \frac{\beta^2 x^2}{\alpha}, \beta x \right\}}$.*

The following concentration bound on the empirical covariance will also be helpful.

**Lemma 3.4** ([27, Fact 3.4]). *There exists a constant $c \geq 1$ such that for any $\Sigma \succeq I$, let $U_1, \ldots, U_n \sim \mathsf{N}(0, \Sigma)$ and $\hat{\Sigma} = \frac{1}{n} \sum_{i \in [n]} U_i U_i^T$, we have $\Pr\left[ \hat{\Sigma} \succeq \left( 1 - c\sqrt{\frac{d + \log(1/\beta)}{n}} \right) \cdot I \right] \geq 1 - \beta$.*

## 3.2 Differential Privacy

**Hockey Stick Divergence.** We also recall the definition of $\epsilon$-*hockey stick divergence* between two distributions $P, Q$: $d_\epsilon(P \,||\, Q) := \int_{y \in \mathrm{supp}(P)} [f_P(y) - e^\epsilon f_Q(y)]_+ dy$, where $[a]_+ := \max\{a, 0\}$. The following standard fact about the hockey stick divergence is often useful in proving DP guarantees of algorithms [39, 31].

**Lemma 3.5.** *For any $\epsilon \geq 0$ and distributions $P, Q$, $d_\epsilon(P \,||\, Q) \leq \Pr_{y \sim P}[f_P(y) > e^\epsilon f_Q(y)]$.*

It will be also useful to keep in mind the "post-processing" property of DP:

**Lemma 3.6.** *For any distributions $P, Q$ and any function $h$, we have $d_\epsilon(h(P) \,||\, h(Q)) \leq d_\epsilon(P \,||\, Q)$.*

**DP under Condition.** Since we will use a "propose-test-release"-style algorithm [17], it will be convenient to use the notion of "DP under condition" together with its composition properties. The particular definition we use below is from [32]; similar notions have been used earlier, e.g., in [17].

**Definition 3.7** (DP under Condition, [32]). Let $\Psi : \mathcal{Y} \to \{0, 1\}$ be a predicate. An algorithm $M : \mathcal{Y} \to \mathcal{O}$ is $(\epsilon, \delta)$-*DP under condition* $\Psi$ for $\epsilon, \delta > 0$ iff, for every $S \subseteq \mathcal{O}$ and every neighboring datasets $Y, Y' \in \mathcal{Y}$ both satisfying $\Psi$, we have $\Pr[\mathcal{M}(Y) \in S] \leq e^\epsilon \cdot \Pr[\mathcal{M}(Y') \in S] + \delta$.

**Lemma 3.8** (Composition for Algorithm with Halting, [32]). *Let $\mathcal{M}_1 : \mathcal{Y} \to \mathcal{O}_1 \cup \{\bot\}, \mathcal{M}_2 : \mathcal{O}_1 \times \mathcal{Y} \to \mathcal{O}_2$ be algorithms. Furthermore, let $\mathcal{M}$ denote the following algorithm: Let $o_1 = \mathcal{M}_1(Y)$ and, if $o_1 = \bot$, then halt and output $\bot$ or else, output $o_2 = \mathcal{M}_2(o_1, Y)$.*

*Let $\Psi$ be any condition such that, if $Y$ does not satisfy $\Psi$, then $\mathcal{M}_1(Y)$ always returns $\bot$. Suppose that $\mathcal{M}_1$ is $(\epsilon_1, \delta_1)$-DP and $\mathcal{M}_2$ is $(\epsilon_2, \delta_2)$-DP under condition $\Psi$. Then, $\mathcal{M}$ is $(\epsilon_1 + \epsilon_2, \delta_1 + \delta_2)$-DP.*

# 4 Gaussian Distribution: Algorithms

**Reduction to the Bounded Mean Case.** As stated earlier, for the cases of known covariance and bounded covariance, we will need a preprocessing step that computes a rough private estimate for the mean. The properties of the reduction are stated below.

**Lemma 4.1.** *Suppose that there is an $\alpha$-accurate $(\epsilon, \delta)$-DP sampler for Gaussian distributions under the assumption that $I \preceq \Sigma \preceq \kappa \cdot I, \|\mu\| \leq R$ with sample complexity $n_{\text{bm-sampler}}(\alpha, R, \epsilon, \delta)$. Then, there exists an $\alpha$-accurate $(\epsilon, \delta)$-DP sampler for Gaussian distributions under the assumption that $I \preceq \Sigma \preceq \kappa \cdot I$ with sample complexity $\tilde{O}\left(\frac{\sqrt{d}}{\epsilon}\right) + n_{\text{bm-sampler}}(\alpha/2, O(\kappa\sqrt{d}), \epsilon/2, \delta/2)$.*

## 4.1 Known Covariance

With the reduction in Lemma 4.1, we may assume that $\|\Sigma^{-1/2}\mu\| \leq R$. When the covariance is known, we give an algorithm with the following guarantees.

**Theorem 4.2.** *Assuming $\Sigma$ is known and $\|\Sigma^{-1/2}\mu\| \leq R$, there is an $\alpha$-accurate $(\epsilon, \delta)$-DP sampler for Gaussian distributions with sample complexity $O\left(\left(R + \sqrt{d + \log\left(\frac{\log(1/\delta)}{\alpha\epsilon}\right)}\right)\frac{\sqrt{\log(1/\delta)}}{\epsilon}\right)$.*

We assume w.l.o.g. that $\Sigma = I$; otherwise, we can consider $X_i' = \Sigma^{-1/2}X_i$. Before we describe the algorithm, note that Theorem 4.2 together with Lemma 4.1 implies Theorem 1.2. As stated earlier, the algorithm (Algorithm 1) is simple: take the average of the truncated input samples and add to it (spherical) Gaussian noise.

*Proof.* Let $C \geq 1$ be the constant from Lemma 3.2, $B = R + 10^4 C \sqrt{d + \log\left(\frac{2\log(2/\delta)}{\alpha\epsilon}\right)}$, and $n = 1 + \lceil 10B\sqrt{\log(2/\delta)}/\epsilon \rceil$. Let $\mathcal{A}$ be Algorithm 1 with $B, n$ as specified and $\sigma = \sqrt{(n-1)/n}$.

**Privacy Analysis.** $\mathcal{A}$ is the Gaussian mechanism with noise multiplier $n\sigma/B \geq 10\sqrt{\log(2/\delta)}/\epsilon$; therefore, $\mathcal{A}$ is $(\epsilon, \delta)$-DP, using [20, Appendix A].

**Accuracy Analysis.** Let $D = \mathsf{N}(\mu, I)$ for some unknown $\mu$. Consider the algorithm $\mathcal{A}'$ where there is no truncation, i.e., $\mathcal{A}'$ simply outputs $Y := Z + \frac{1}{n}\sum_{i\in[n]} X_i$. Via Lemma 3.2 and a union bound, the truncation is not applied anyway in $\mathcal{A}$ (i.e., $X_i = X_i^{\text{trunc}}, \forall i \in [n]$) with probability at least $1 - \alpha$. Therefore, $d_{\text{tv}}(Q_{\mathcal{A},D}, Q_{\mathcal{A}',D}) \leq \alpha$. Note that $\mathcal{A}'$ just outputs $Y := Z + \frac{1}{n}\sum_{i\in[n]} X_i$, so we have $Y \sim \mathsf{N}(\mu, I) = D$, i.e., $Q_{\mathcal{A}',D} = D$. Combining these bounds yields $d_{\text{tv}}(Q_{\mathcal{A},D}, D) \leq \alpha$. □

## 4.2 Unknown Bounded Covariance

We now move on to the case where both $\mu, \Sigma$ are unknown but under the assumption $I \preceq \Sigma \preceq \kappa \cdot I$. The main result of this section is stated below[7]. Again, note that Theorem 4.3 and Lemma 4.1 immediately yield Theorem 1.4.

**Theorem 4.3.** *Assuming $I \preceq \Sigma \preceq \kappa \cdot I$ for some $\kappa > 0$ and $\|\mu\| \leq R$, there is an $\alpha$-accurate $(\epsilon, \delta)$-DP sampler for Gaussian distributions with sample complexity $O\left(\left(R^2 + \kappa^2\left(d + \log\left(\frac{\log(1/\delta)}{\alpha\epsilon}\right)\right)\right) \cdot \frac{\log(1/\delta)}{\epsilon}\right)$.*

As stated in the overview, the main challenge lies in the privacy analysis. It will be proved in three steps. First, in Section 4.2.1, we will show that if we add noise that is drawn from projection of random unit vector to the first $M$ coordinates to a low-($\ell_2$-)sensitivity function, then it is DP. Then, in Section 4.2.2, we use this to show that adding noise of the form $\sum_{i\in[n]} a[i] \cdot w_i$ (where $a \sim \mathsf{Uni}_n^{\mathbb{S}}$) to a low-sensitivity function also suffices for privacy as long as the smallest eigenvalue of $\sum_{i\in[n]} w_i w_i^T$ is sufficiently large. Here $w_1, \ldots, w_n$ are assumed to be fixed vectors given beforehand. Note that the noises discussed so far are input *independent*. Finally, in Appendix B.4, we relate this to the

---

[7]The dependence on $\kappa$ can be reduced to polylog$(1/\kappa)$ by applying the private preconditioning in [27], although this will increase the dependence on $d$ to $d^{1.5}$; it is an interesting question if this increase in the dependence on $d$ can be avoided.

privacy of our algorithm—which is more intricate as these $w_i$'s are now $X_i$'s, i.e., the noise is input dependent. The accuracy analysis, which is simpler than the privacy analysis, is also in Appendix B.4.

### 4.2.1 Noise via Random Unit Vector Projection

We first show that adding noise drawn from $\mathsf{Uni}^{\mathbb{S}}_{M,N}$ to a function with low $\ell_2$-sensitivity ensures DP.

**Lemma 4.4.** *Let $M, N$ be any positive integers such that $M \geq \max\left\{2N, \frac{10^4 \log(10/\delta)}{c\epsilon}\right\}$, where $c$ is the constant from Lemma 3.3. Furthermore, let $\tau \in (0, 0.01)$ be such that $\tau \leq 10\sqrt{\frac{\log(10/\delta)}{cM}}$. Then, for any vector $v$ such that $\|v\| \leq \tau$, we have $d_\epsilon(\mathsf{Uni}^{\mathbb{S}}_{M,N} \,\|\, \mathsf{Uni}^{\mathbb{S}}_{M,N} + v) \leq \delta$.*

The proof of this lemma proceeds similarly to those of independent noise addition (e.g., the Gaussian mechanism): using Lemma 3.5, it suffices to bound the probability that the privacy loss is large. We achieve such a bound via concentration properties of the beta distribution.

### 4.2.2 Noise via Spherical Linear Combinations

Next, we relate the previous lemma to privacy in the case where the noise added is of the form $\sum_{i \in [q]} a[i] \cdot w_i$, where $w_1, \dots, w_q$ are fixed beforehand and $a \sim \mathsf{Uni}^{\mathbb{S}}_q$. The guarantee of such a noise is stated below. The proof is based on a reduction argument to Lemma 4.4. For notational convenience, for every matrix $W \in \mathbb{R}^{d \times q}$, let $\mathsf{Uni}^{\mathbb{S}}_W$ be the distribution of $Wa$ where $a \sim \mathsf{Uni}^{\mathbb{S}}_q$.

**Lemma 4.5.** *Let $q, d \in \mathbb{N}$ and $\tau_0 \in (0, 1)$ be such that $q \geq \max\left\{2d, \frac{10^4 \log(10/\delta)}{c\epsilon}\right\}$ and $\tau_0 \leq 5\sqrt{\log(10/\delta)/c}$. For any $w_1, \dots, w_q \in \mathbb{R}^d$ such that $\frac{1}{q}\left(\sum_{i=1}^q w_i w_i^T\right) \succeq 0.5I$ and any $v \in \mathbb{R}^d$ such that $\|v\| \leq \tau_0$, we have $d_\epsilon(\mathsf{Uni}^{\mathbb{S}}_W \,\|\, \mathsf{Uni}^{\mathbb{S}}_W + v) \leq \delta$, where $w_1, \dots, w_q$ are the columns of $W \in \mathbb{R}^{d \times q}$.*

*Proof.* Let $W^+ \in \mathbb{R}^{d \times q}$ denote the pseudoinverse of $W$. Note that $WW^T = \sum_{i=1}^q w_i w_i^T \succeq 0.5qI$. Therefore, all singular values of $W$ are at least $\sqrt{0.5q}$. This means that all singular values of $W^+$ are at most $1/\sqrt{0.5q}$. This in turn implies that $\|W^+ v\| \leq \|v\|/\sqrt{0.5q} \leq \tau_0/\sqrt{0.5q} =: \tau$.

Since $W^T W \succeq 0.5qI$, $W^T$ is full rank and thus $\dim(\mathrm{im}(W^+)) = d$. Since $d < q$, pick any orthonormal basis $b_1, \dots, b_d \in \mathbb{R}^q$ for $\mathrm{im}(W^+)$ and let $\Pi : \mathbb{R}^q \to \mathbb{R}^d$ be such that $\Pi(z) = (\langle z, b_1\rangle, \dots, \langle z, b_d\rangle)$.

Furthermore, let $h : \mathbb{R}^d \to \mathbb{R}^d$ be defined as $h(x) := \sum_{i \in [d]} x[i] \cdot b_i$. Observe that, for any $a \in \mathbb{R}^q$, we have $a_1 u_1 + \cdots + a_q u_q = h(\Pi(a))$ and $a_1 u_1 + \cdots + a_q u_q + v = h(\Pi(a) + \Pi(W^+ v))$. Thus,

$$d_\epsilon(\mathsf{Uni}^{\mathbb{S}}_W \,\|\, \mathsf{Uni}^{\mathbb{S}}_W + v) = d_\epsilon(h(\Pi(\mathsf{Uni}^{\mathbb{S}}_q)) \,\|\, h(\Pi(\mathsf{Uni}^{\mathbb{S}}_q) + \Pi(W^+ v)))$$

$$\overset{\text{Lemma 3.6}}{\leq} d_\epsilon(\Pi(\mathsf{Uni}^{\mathbb{S}}_q) \,\|\, \Pi(\mathsf{Uni}^{\mathbb{S}}_q) + \Pi(W^+ v)).$$

Note that $\Pi(\mathsf{Uni}^{\mathbb{S}}_q) \sim \mathsf{Uni}^{\mathbb{S}}_{q,d}$. So, Theorem 4.4 implies $d_\epsilon(\Pi(\mathsf{Uni}^{\mathbb{S}}_q) \,\|\, \Pi(\mathsf{Uni}^{\mathbb{S}}_q) + \Pi(W^+ v)) \leq \delta$. $\quad\square$

## 5 Discussion and Open Questions

In this work, we give several algorithms for DP sampling. For Gaussian distributions, we demonstrate that the dependence on the accuracy parameter $\alpha$ in the sample complexity can be only polylogarithmic in comparison to the polynomial dependence necessary in DP learning. Furthermore, we also reduce the dependence on $d$ in the case of known variance and unknown bounded variance; in both cases, we show this dependence (of $\sqrt{d}$ and $d$ respectively) to be essentially tight. An immediate open question is whether the dependence of $d^{1.5}$ in the case of unknown unbounded variance is tight. Lower bounds for Gaussian sampling in terms of $\alpha$ is also an interesting question. Another—more general—direction is to extend the results to other families of distributions. In the case of DP learning, this has been explored by several works. Our algorithms very specifically use the property of Gaussian distributions (namely that a linear combination of Gaussians is another Gaussian); other distribution families might need different techniques.

**Acknowledgment.** We thank Shyam Narayanan for pointing us to [9], which improves dependency on $d$ in the sample complexity for sampling Gaussians with unknown (unbounded) covariance from $d^2$ to $d^{1.5}$.

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

# A  Additional Preliminaries

We consider a generalization of the truncation (1) where there is a weight vector $w \in \mathbb{R}^d_{\geq 0}$, and the truncation is w.r.t. $w$:

$$\text{trunc}^p_{B,w}(X) := \begin{cases} X & \text{if } \|X \circ w\|_p \leq B, \\ X \cdot B/\|X \circ w\|_p & \text{if } \|X \circ w\|_p > B, \end{cases}$$

where $\circ$ denotes the element-wise product. Clearly $\text{trunc}^p_B = \text{trunc}^p_{B,w}$ when $w$ is the all-ones vector.

For $a, b \in \mathbb{R}$ where $a \leq b$, we also use $\text{clip}_{(a,b)} : \mathbb{R} \to \mathbb{R}$ to denote the function:

$$\text{clip}_{(a,b)}(x) = \max\{a, \min\{b, x\}\}.$$

## A.1  Total Variation Distance between Gaussians

For our proofs, it will be useful to state the total variation distance bounds for Gaussian distributions. We will use the following notation:

**Definition A.1.** For any matrix $A \in \mathbb{R}^{d \times d}$ and positive semi-definite matrix $\Sigma \in \mathbb{R}^{d \times d}$, we write $\|A\|_\Sigma := \|\Sigma^{-1/2} A \Sigma^{-1/2}\|_F$, where $\|\cdot\|_F$ denotes the Frobenius norm.

It is well known that when $\|\Sigma - \hat{\Sigma}\|_\Sigma \leq O(1)$, the total variation distance between Gaussian with covariance matrices $\Sigma$ and $\hat{\Sigma}$ and the same mean is $\Theta(\|\Sigma - \hat{\Sigma}\|_\Sigma)$, as stated more formally below.

**Lemma A.2** ([13]). *There exists a constant $C \geq 1$ such that the following holds. For any $\Sigma, \hat{\Sigma}$, if $d_{\text{tv}}(\mathsf{N}(0, \Sigma), \mathsf{N}(0, \hat{\Sigma})) \leq 0.001$, then $\|\Sigma - \hat{\Sigma}\|_\Sigma \leq C \cdot d_{\text{tv}}(\mathsf{N}(0, \Sigma), \mathsf{N}(0, \hat{\Sigma}))$.*

**Lemma A.3** (e.g., [14]). *There exists a constant $C > 0$ such that the following holds. For any $\Sigma, \hat{\Sigma} \in \mathbb{R}^{d \times d}$, we have $d_{\text{tv}}(\mathsf{N}(0, \Sigma), \mathsf{N}(0, \hat{\Sigma})) \leq C \cdot \|\Sigma - \hat{\Sigma}\|_\Sigma$.*

## A.2  Differential Privacy

### A.2.1  Amplification by Subsampling

Suppose that we have as an input $N$ samples, and we draw $n < N$ subsamples (without replacement) randomly from these $N$ samples and run an $(\epsilon, \delta)$-DP algorithm on these $n$ subsamples. Then, it is known that this results in an $(\epsilon', \delta')$-DP algorithm for $\epsilon' < \epsilon, \delta' < \delta$. Such a phenomenon is called *amplification by subsampling* and is often used in DP learning [e.g., 1]. We will use the following version of this for our lower bound proof.

**Theorem A.4** (Amplification by Subsampling, [6]). *Suppose that $\mathcal{A}$ is an $(\epsilon, \delta)$-DP algorithm that takes in $n$ samples. Then, the algorithm that draws $n$ subsamples randomly without replacement out of $N$ samples and runs $\mathcal{A}$ on the $n$ subsamples is $(\epsilon', \delta')$-DP where*

$$\epsilon' = \ln(1 + (n/N)(e^\epsilon - 1)), \qquad \delta' = (n/N) \cdot \delta.$$

## A.3  Advanced Composition

We will also use the following *advanced composition* of DP.

**Theorem A.5** (Advanced Composition, [19]). *Let $\epsilon_0 > 0$ and $\delta_0, \delta_1 \in (0, 1/2]$, and $\mathcal{A}$ be any $(\epsilon_0, \delta_0)$-DP algorithm. Then, the algorithm that runs $\mathcal{A}$ a total of $k$ times is $(\epsilon_1, k\delta_0 + \delta_1)$-DP where*

$$\epsilon_1 = \left( \sqrt{2k \ln(1/\delta_1)} + k(e^{\epsilon_0} - 1) \right) \epsilon_0.$$

## A.4  Lower Bounds for Mean and Covariance Estimations

As stated in our proof overview (Section 2), our lower bounds for Gaussian samplers (Theorems 1.3 and 1.5) are via reductions from previously known lower bounds for mean and covariance estimations. We state them below, starting with the mean estimation lower bound (aka "fingerprinting for Gaussian") result due to [27]

**Theorem A.6** ([27])**.** *For any $\epsilon, \delta \in (0, 1]$ such that $\delta \leq O\left(\frac{\sqrt{d}}{n\sqrt{\log(n/d)}}\right)$, if an $(\epsilon, \delta)$-DP algorithm $\mathcal{M}$ that can take in $N$ samples from $D = \mathsf{N}(\mu, I)$ where[8] $-1 \leq \mu \leq 1$ and output an estimate $\hat{\mu} \in \mathbb{R}^d$ that satisfies*

$$\mathbb{E}_{\mathbf{X} \sim D^N, \hat{\mu} \sim \mathcal{M}(\mathbf{X})}[\|\hat{\mu} - \hat{\mu}\|_2^2] \leq \gamma^2 \leq d/6,$$

*then $N \geq \Omega\left(\frac{d}{\gamma\epsilon}\right)$.*

Next is a similar lower bound but for covariance estimation. Notice that the lower bound on the sample complexity below is $\Omega(d^2/\epsilon^2)$ whereas the lower bound above is only $\Omega(d/\epsilon)$ (for constant accuracy parameter $\gamma$).

**Theorem A.7** ([28])**.** *There exists a small constant $\xi > 0$ such that the following holds: for any $\epsilon, \delta \in (0, 1]$ such that $\delta \leq O\left(\min\left\{1/n, d^2/(n \log n)\right\}\right)$, if an $(\epsilon, \delta)$-DP algorithm $\mathcal{M}$ that can take in $N$ samples from $D = \mathsf{N}(0, \Sigma)$ where $I \preceq \Sigma \preceq 2I$ and output an estimate $\hat{\Sigma} \in \mathbb{R}^{d \times d}$ that satisfies*

$$\mathbb{E}_{\mathbf{X} \sim D^N, \hat{\Sigma} \sim \mathcal{M}(\mathbf{X})}[\|\hat{\Sigma} - \Sigma\|_{\Sigma}^2] \leq \xi^2,$$

*then $N \geq \Omega\left(d^2/\epsilon\right)$.*

### A.5 Agnostic Learner

In the main body of the paper, we considered the learner where the samples are drawn from $D$ which belongs to a certain class $\mathcal{D}$. For our lower bound proofs, it will be convenient to consider the *agnostic* setting where $D$ is not assumed to be from $\mathcal{D}$. The definition and accuracy guarantee of agnostic learner is given below. We remark that the definition coincides with the learner definition given earlier if we assume that $D \in \mathcal{D}$. (Note that the constant 3 is required for Theorem A.8.)

**Agnostic Learning.** Let $\mathcal{D}$ be a class of distribution. An algorithm $\mathcal{A}$, which takes in $n$ samples and output a distribution $\hat{P} \in \mathcal{D}$, is said to be an $(\alpha, \beta)$-*accurate agnostic learner* for a class $\mathcal{D}$ of distributions iff, for any distribution $P$, we have

$$\Pr_{\mathbf{X} \sim P^n, \hat{P} \leftarrow \mathcal{A}(\mathbf{X})}\left[d_{\mathrm{tv}}(\hat{P}, P) \leq 3 \cdot \min_{P \in \mathcal{D}} d_{\mathrm{tv}}(D, P) + \alpha\right] \geq 1 - \beta.$$

We will use the following (well-known) result that the class of centered[9] Gaussian distributions can be learned with sample complexity $O(d^2/\gamma^2)$. For a full proof, see e.g., [3].

**Theorem A.8.** *For any $\gamma, \beta \in (0, 1/2]$, there exists a $(\gamma, \beta)$-accurate agnostic learner for centered Gaussian distributions in $d$ dimensions with sample complexity*

$$O\left(\frac{d^2 + \sqrt{\log(1/\beta)}}{\gamma^2}\right).$$

## B Proofs for Section 4

### B.1 Reduction to the Bounded Mean Case

The reduction will be done by applying a so-called "densest ball" algorithm; this is an algorithm that, when there is a ball of radius $r$ that contains the majority of the input, can find a ball of radius $O(r)$ containing the majority of the input. There are many algorithms known for this problem [23, 40, 33] that give similar guarantees[10]. We state such a guarantee below.

---

[8]We write $-1 \leq \mu \leq 1$ as a shorthand for $-1 \leq \mu[j] \leq 1$ for all $j \in [d]$.

[9]Recall that centered simply means that $\mu = 0$.

[10]We note that our problem is in fact easier than those considered in the aforementioned work as we assume that $\Sigma \preceq \kappa \cdot I$ and the problem can be solved "coordinate-by-coordinate". This means that many other algorithms e.g., truncated Laplace-based histogram [12] also work and give similar bounds. However, we note that some other algorithms [e.g., 7] require prior bounds on $\|\mu\|$, whereas Theorem B.1 does not require any such bound.

**Theorem B.1** (e.g., [23]). *For $n_{\text{dens}}(d, \beta, \epsilon, \delta) = \tilde{\Theta}\left(\frac{\sqrt{d}}{\epsilon}\right)$, there exists an $(\epsilon, \delta)$-DP algorithm $\mathcal{A}_{\text{dens}}$ that takes in $X_1, \ldots, X_n \in \mathbb{R}^d$ together with a radius parameter $r > 0$ such that: if there is an $r$-radius ball containing at least $2/3$ fraction of the input points, then with probability $1 - \beta$, it outputs a ball of radius $O(r)$ that contains at least half of the input points.*

We can now give a reduction of an arbitrary mean case to the bounded mean case by first using Theorem B.1 to obtain a rough estimate of the mean and then shifting the remaining samples accordingly.

*Proof.* The new sampler works as follows:

- Let $n_1 := \max\{n_{\text{dens}}(d, \alpha/4, \epsilon/2, \delta/2), 100 \log(4/\alpha)\}, r = 10C\kappa\sqrt{d}$ where $C$ is the constant from Lemma 3.2, and $n_2 := n_{\text{bm-sampler}}(\alpha/2, r, \epsilon/2, \delta/2)$.
- *Rough Mean Estimator Stage:*
  - Run $(\epsilon/2, \delta/2)$-DP $\mathcal{A}_{\text{dens}}$ from Theorem B.1 on $n_1$ samples (with $r$ as specified above) to get a ball centered at $c \in \mathbb{R}^d$.
- *Sampling Stage:*
  - Draw $n_2$ additional samples $X_1, \ldots, X_{n_2}$.
  - Run $(\epsilon/2, \delta/2)$-DP $(\alpha/2)$-accurate $\mathcal{A}_{\text{bm-sampler}}$ with $R = r$ on $(X_1 - c), \ldots, (X_{n_2} - c)$; let $Y$ be its output.
  - Output $Y + c$.

**Privacy Analysis.** Basic composition implies that the sampler is $(\epsilon, \delta)$-DP as desired.

**Accuracy Analysis.** Since $\Sigma \preceq \kappa I$, Lemma 3.2 implies that $\Pr_{X \sim \mathsf{N}(\mu, \Sigma)}[\|X - \mu\|_2 \geq r] \leq 0.1$. Thus, using standard concentration inequality (e.g., Theorem E.1) and since $n_1 \geq 100 \log(4/\alpha)$, the probability that at least $2/3$ of the $n_1$ points sampled in the first stage contains in the ball $B(\mu, r)$ is at least $1 - \alpha/4$. The guarantee of the densest ball algorithm then implies that with probability $1 - \alpha/2$, we have $\|\mu - c\|_2 \leq r$.

Let us consider a fixed $c$ and consider the output distribution $\mathcal{D}_Y^c$ of $Y$ in the sampling stage and $\mathcal{D}_{Y+c}^c$ of the final output $Y + c$. Notice that $(X_1 - c), \ldots, (X_{n_2} - c) \sim \mathsf{N}(\mu - c, \Sigma)$. Consequently, when $\|\mu - c\|_2 \leq r$, we may apply the guarantee of $\mathcal{A}_{\text{bm-sampler}}$ to yield

$$\alpha/2 \geq d_{\text{tv}}(\mathcal{D}_Y^c, \mathsf{N}(\mu - c, \Sigma)) = d_{\text{tv}}(\mathcal{D}_{Y+c}^c, \mathsf{N}(\mu, \Sigma)).$$

Combining these arguments, the sampler is $\alpha$-accurate as desired. $\qquad\square$

## B.2 Unknown Bounded Covariance

*Proof of Lemma 4.4.* From Lemma 3.5, we have

$$d_\epsilon(\mathsf{Uni}_{M,N}^{\mathbb{S}} \| \mathsf{Uni}_{M,N}^{\mathbb{S}} + v) \leq \Pr_{z \sim \mathsf{Uni}_{M,N}^{\mathbb{S}}}[f_{\mathsf{Uni}_{M,N}^{\mathbb{S}}}(z) > e^\epsilon \cdot f_{\mathsf{Uni}_{M,N}^{\mathbb{S}}}(z - v)].$$

For convenience, let us also define $\eta = \tau \cdot \left(10\sqrt{\frac{\log(10/\delta)}{cM}}\right)$. Note that $\eta \leq 0.01$.

For $z$ such that $\|z\| \leq 0.9, |\langle z, v \rangle| \leq \eta$, we also have $\|z - v\|^2 \leq 0.9^2 + 0.02 + 0.0001 < 0.9$ and thus

$$\ln \frac{\mathsf{Uni}_{M,N}^{\mathbb{S}}(z)}{\mathsf{Uni}_{M,N}^{\mathbb{S}}(z - v)} \overset{(2)}{=} \left(\frac{M - N}{2} - 1\right) \ln \left(\frac{1 - \|z\|^2}{1 - \|z - v\|^2}\right)$$

$$\leq \left(\frac{M - N}{2} - 1\right) \left(\frac{1 - \|z\|^2}{1 - \|z - v\|^2} - 1\right)$$

$$\leq \left(\frac{M - N}{2} - 1\right) \left(\frac{\|v\|^2 - 2\langle z, v \rangle}{1 - \|z - v\|^2}\right)$$

$$\leq M\left(10(\tau^2 + 2\eta)\right)$$

$$\leq \epsilon,$$

where the last inequality follows from our choice of parameters (i.e., $M \geq \frac{10^4 \log(10/\delta)}{c\epsilon}$).

Combining the previous two bounds, we have

$$d_\epsilon(\mathsf{Uni}^{\mathbb{S}}_{M,N} \,\|\, \mathsf{Uni}^{\mathbb{S}}_{M,N} + v) \leq \Pr_{z \sim \mathsf{Uni}^{\mathbb{S}}_{M,N}} [\|z\| > 0.9 \,\vee\, \langle z, v \rangle > \eta]$$

$$\leq \Pr_{z \sim \mathsf{Uni}^{\mathbb{S}}_{M,N}} [\|z\| > 0.9] + \Pr_{z \sim \mathsf{Uni}^{\mathbb{S}}_{M,N}} [\langle z, v \rangle > \eta]$$

$$\leq \Pr_{z \sim \mathsf{Uni}^{\mathbb{S}}_{M,N}} [\|z\| > 0.9] + \Pr_{z \sim \mathsf{Uni}^{\mathbb{S}}_{M,N}} \left[ \left\langle z, \frac{v}{\|v\|} \right\rangle > 10\sqrt{\frac{\log(10/\delta)}{cM}} \right].$$

Next, recall that when $z \sim \mathsf{Uni}^{\mathbb{S}}_{M,N}$, $\|z\|^2 \sim \mathsf{Beta}(N/2, (M-N)/2)$.[11] Thus, applying Lemma 3.3 and using the assumption that $N \leq M/2$, we can conclude that

$$\Pr_{z \sim \mathsf{Uni}^{\mathbb{S}}_{M,N}} [\|z\| > 0.9] = \Pr_{Z \sim \mathsf{Beta}(N/2,(M-N)/2)} [Z^2 > 0.81] \leq 2\exp(-c \cdot 0.01 M) \leq \delta/2,$$

where the first inequality is from Lemma 3.3 and the last inequality follows from our assumption that $M \geq 100 c \log(10/\delta)$.

Recall also that when $z \sim \mathsf{Uni}^{\mathbb{S}}_{M,N}$, $\left| \left\langle z, \frac{v}{\|v\|} \right\rangle \right|^2 \sim \mathsf{Beta}(1/2, (M-1)/2)$. Again, applying Lemma 3.3 yields

$$\Pr_{z \sim \mathsf{Uni}^{\mathbb{S}}_{M,N}} \left[ \left\langle z, \frac{v}{\|v\|} \right\rangle > 10\sqrt{\frac{\log(10/\delta)}{cM}} \right] \leq \Pr_{Z \sim \mathsf{Beta}(1/2,(M-1)/2)} \left[ Z^2 > \frac{100\log(10/\delta)}{cM} \right] \leq \delta/2,$$

where the last inequality again follows from our choice of parameters.

Combining the three preceding inequalities, we can conclude that $d_\epsilon(\mathsf{Uni}^{\mathbb{S}}_{M,N} \,\|\, \mathsf{Uni}^{\mathbb{S}}_{M,N} + v) \leq \delta$ as desired. □

### B.3 Unknown Covariance

Throughout this section, we assume the standard assumption that the covariance matrix $\Sigma$ is full rank. It is simple to extend the result to the more general case using an algorithm of [4], which can identify the span of $\Sigma$ with probability one, using $O(d\log(1/\delta)/\epsilon)$ samples.

As stated earlier, we will use DP preconditioner from [9]. We say that an algorithm is an $(\alpha, \beta)$-*accurate preconditioner for Gaussian distributions* iff, when the input $\mathbf{X}$ is drawn i.i.d. from $\mathsf{N}(\mu, \Sigma)$ (for any $\mu \in \mathcal{R}^d, \Sigma \in \mathcal{R}^{d \times d}$), the algorithm outputs $\hat{\mu}, \hat{\Sigma}$ that satisfies the following with probability at least $1 - \beta$: $\|\hat{\Sigma}^{-1}(\hat{\mu} - \mu)\|_2 \leq \alpha$ and[12] $\|\Sigma^{-1/2}\hat{\Sigma}\Sigma^{-1/2} - I\|_2 \leq \alpha$. Below we provide a generic reduction from the unbounded covariance case to the bounded covariance case using preconditioner.

**Lemma B.2.** *Suppose that the following hold:*

- *There exists an $(\alpha, \beta)$-accurate $(\epsilon, \delta)$-DP preconditioner $\mathcal{A}_{\mathrm{precnd}}$ for Gaussian distributions with sample complexity $n_{\mathrm{precnd}}(\alpha, \beta, \epsilon, \delta)$.*
- *There exists an $\alpha$-accurate $(\epsilon, \delta)$-DP sampler $\mathcal{A}_{\mathrm{bc\text{-}sampler}}$ for Gaussian distributions with $I \preceq \Sigma \preceq \kappa \cdot I$ with sample complexity $n_{\mathrm{bc\text{-}sampler}}(\kappa, \alpha, \epsilon, \delta)$.*

*Then, there exists an $\alpha$-accurate $(\epsilon, \delta)$-DP sampler for Gaussian (without any assumption) with sample complexity*

$$n_{\mathrm{precnd}}(0.001, \alpha/2, \epsilon/2, \delta/2) + n_{\mathrm{bc\text{-}sampler}}(4, \alpha/2, \epsilon/2, \delta/2).$$

*Proof of Lemma B.2.* The sampler is as follows:

- Let $n_1 := n_{\mathrm{precnd}}(0.001, \alpha/2, \epsilon/2, \delta/2)$ and $n_2 := n_{\mathrm{bc\text{-}sampler}}(4, \alpha/2, \epsilon/2, \delta/2)$.
- *Estimation Stage:*

---

[11]See, e.g., `https://en.wikipedia.org/wiki/Beta_distribution`

[12]The guarantee here is in the spectral norm, not Frobenius norm. However, this suffices for our purposes.

- Run $(\epsilon/2, \delta/2)$-DP $(0.001, \alpha/2)$-accurate $\mathcal{A}_{\text{precnd}}$ on $n_1$ samples to get $\hat{\mu}, \hat{\Sigma}$.
- *Sampling Stage:*
  - Draw $n_2$ additional samples $X_1, \ldots, X_{n_2}$.
  - Run $(\epsilon/2, \delta/2)$-DP $(\alpha/2)$-accurate $\mathcal{A}_{\text{bc-sampler}}$ on $2\hat{\Sigma}^{-1/2}(X_1 - \hat{\mu}), \ldots, 2\hat{\Sigma}^{-1/2}(X_{n_2} - \hat{\mu})$; let $Y$ be its output.
  - Output $0.5\hat{\Sigma}^{1/2}Y + \hat{\mu}$.

Basic composition implies that the sampler is $(\epsilon, \delta)$-DP.

To see the accuracy guarantee, the guarantee of the learner implies that with probability $1 - \alpha/2$, we have $\|\hat{\Sigma}^{-1}(\hat{\mu} - \mu)\|_2 \leq 0.001$ and $\|\Sigma^{-1/2}\hat{\Sigma}\Sigma^{-1/2} - I\|_2 \leq 0.001$. The latter implies that all eigenvalues of $\hat{\Sigma}^{1/2}\Sigma^{-1/2}$ lie in [0.998, 1.002]; in turn, this implies that the eigenvalues of $\hat{\Sigma}^{-1/2}\Sigma^{1/2}$ all lie in [0.996, 1.004]. Finally, this implies that $\|\Sigma^{1/2}\hat{\Sigma}^{-1}\Sigma^{1/2} - I\|_2 \leq 0.01$.

Let us consider a fixed $\hat{\Sigma}, \hat{\mu}$ and consider the output distribution of the sampling stage. When the above inequalities hold, the guarantee of $\mathcal{A}_{\text{bc-sampler}}$ implies that

$$\alpha/2 \geq d_{\text{tv}}(Y, \mathsf{N}(2\hat{\Sigma}^{-1/2}(\mu - \hat{\mu}), 4\Sigma^{1/2}\hat{\Sigma}^{-1}\Sigma^{1/2})) = d_{\text{tv}}(0.5\hat{\Sigma}^{1/2}Y + \hat{\mu}, \mathsf{N}(\mu, \Sigma)).$$

Combining the above arguments, the sampler is $\alpha$-accurate as desired. □

Combining the above reduction (Lemma B.2) together with a known result on DP preconditioner of Gaussians (Theorem B.3 below), we immediately arrive at Theorem 1.6.

**Theorem B.3** ([8]). *There is an $(\alpha, \beta)$-accurate $(\epsilon, \delta)$-DP preconditioner for Gaussian distributions with sample complexity $O\left(\frac{d}{\alpha^2} + \frac{d^{3/2}}{\alpha\epsilon} \cdot \text{polylog}\left(\frac{d}{\alpha\beta\epsilon\delta}\right)\right)$.*

### B.4 Proof of Theorem 4.3

We use Algorithm 2 with the following parameters: $B = R + 10^4 \kappa \sqrt{d + \log\left(\frac{2\log(2/\delta)}{\alpha\epsilon}\right)}, \Delta = 2B^2$, and $n_1 = n_2 = \left\lceil 10^4 C^2 \left(B^2 \cdot \frac{\log(10/\delta)}{c\epsilon}\right)\right\rceil$, where $c, C$ are the constants from Lemmas 3.3 and 3.4.

**Privacy Analysis.** To analyze the privacy constraint, we first notice that the sensitivity of $\lambda_{\min}\left(\sum_{i \in [n_2]} U_i U_i^T\right)$ is at most $\Delta = (\sqrt{2} \cdot B)^2$ and Lemma 3.1 ensures that the check, which only uses $\lambda_{\min}\left(\sum_{i \in [n_2]} U_i U_i^T\right) + r$, is $(\epsilon/2, \delta/2)$-DP. Passing this check ensures that

$$\lambda_{\min} \cdot \left(\sum_{i \in [n_2]} U_i U_i^T\right) \geq 0.75n_2. \tag{3}$$

From Lemma 3.8, it suffices to show that the output of the algorithm is DP, under (3). Let $\mathbf{X} = X_1, \ldots, X_{n_1+2n_2}$.

To do this, it will be more convenient to define $V_i = \sqrt{1 - \frac{1}{n_1}} \cdot U_i$. Condition (3) implies that

$$\lambda_{\min}\left(\sum_{i \in [n_2]} V_i V_i^T\right) \geq 0.5n_2 + \Delta. \tag{4}$$

Let $\mathcal{M}(\mathbf{X}) = \frac{1}{n_1}\left(\sum_{i \in [n_1]} X_i^{\text{trunc}}\right) + \left(\sum_{i \in [n_2]} a[i] \cdot V_i\right)$. Consider any neighboring inputs $\mathbf{X}$ and $\tilde{\mathbf{X}}$ that satisfy the condition. There are two cases, based on where they differ.

(i) *Case I:* $\mathbf{X}, \tilde{\mathbf{X}}$ differ on the first $n_1$ samples. Let $v = \frac{1}{n_1}\sum_{i \in [n_1]}(\tilde{X}_i^{\text{trunc}} - X_i)$; notice that $\|v\| \leq \frac{2B}{n_1}$. Furthermore, (4) implies that $\frac{1}{n_2}\left(\sum_{i \in [n_2]} V_i V_i^T\right) \succeq 0.5I$. Letting $W$ denote the matrix whose columns are $V_1, \ldots, V_{n_2}$, $d_{\frac{\epsilon}{2}}\left(\mathcal{M}(\mathbf{X}) \| \mathcal{M}(\tilde{\mathbf{X}})\right) = d_{\frac{\epsilon}{2}}\left(\mathsf{Uni}_W^\mathbb{S} \| \mathsf{Uni}_W^\mathbb{S} + v\right) \leq \delta/2$, where the inequality follows from Lemma 4.5.

(ii) *Case II:* $\mathbf{X}, \tilde{\mathbf{X}}$ differ on the last $n_2$ samples; assume w.l.o.g. $X_{n_1+2n_2} \neq \tilde{X}_{n_1+2n_2}$. Let $w = V'_{n_2} - V_{n_2}$ and $W$ denote the matrix whose columns are $V_1, \ldots, V_{n_2-1}$. Furthermore, let $a, a'$ be independent samples from $\mathsf{Uni}_q^{\mathbb{S}}$.

$$
d_{\frac{\epsilon}{2}}\left(\mathcal{M}(\mathbf{X}) \,\|\, \mathcal{M}(\tilde{\mathbf{X}})\right) = d_{\frac{\epsilon}{2}}\left(a[n_2] \cdot V_{n_2} + \sum_{i \in [n_2-1]} a[i] \cdot V_i \,\Bigg\|\, a'[n_2] \cdot V'_{n_2} + \sum_{i \in [n_2-1]} a'[i] \cdot V_i\right)
$$

$$
\leq \int_y d_{\frac{\epsilon}{2}}\left(\mathsf{Uni}_W^{\mathbb{S}} \,\Bigg\|\, \frac{y \cdot w}{\sqrt{1-y^2}} + \mathsf{Uni}_W^{\mathbb{S}}\right)\mathsf{Uni}_{n_2,1}^{\mathbb{S}}(y)dy, \tag{5}
$$

where the inequality follows from the coupling $a[n_2] = a'[n_2] = y$, $a[n_2] \sim \mathsf{Uni}_{n_2,1}^{\mathbb{S}}$ and the observation that $\left(\frac{a[1]}{\sqrt{1-y^2}}, \ldots, \frac{a[n_2-1]}{\sqrt{1-y^2}}\right)$ and $\left(\frac{a'[1]}{\sqrt{1-y^2}}, \ldots, \frac{a'[n_2-1]}{\sqrt{1-y^2}}\right)$ are independently distributed as $\mathsf{Uni}_{n_2-1}^{\mathbb{S}}$.

When $|y| \leq \sqrt{\frac{10\log(10/\delta)}{cn_2}}$, we have $\left\|\frac{y \cdot w}{\sqrt{1-y^2}}\right\| \leq \sqrt{\frac{20\log(10/\delta)}{cn_2}} \cdot \sqrt{2} \cdot B \leq 20\sqrt{\frac{\log(10/\delta)}{c}}$. Furthermore, (4) implies that $\frac{1}{n_2}\left(\sum_{i \in [n_2-1]} V_i V_i^T\right) \succeq 0.5I$. Thus, Lemma 4.5 implies

$$
d_{\epsilon/2}\left(\mathsf{Uni}_W^{\mathbb{S}} \,\Bigg\|\, \frac{y \cdot w}{\sqrt{1-y^2}} + \mathsf{Uni}_W^{\mathbb{S}}\right) \leq \delta/4.
$$

Combining this with (5), we arrive at $d_{\epsilon/2}\left(\mathcal{M}(\mathbf{X}) \,\|\, \mathcal{M}(\tilde{\mathbf{X}})\right) \leq \delta/4 + \Pr_{y \sim \mathsf{Uni}_{n_2,1}^{\mathbb{S}}}\left[|y| > \sqrt{\frac{10\log(10/\delta)}{cn_2}}\right]$. Since $y^2 \sim \mathsf{Beta}(1/2, (n_2-1)/2)$, we may apply the tail bound (Lemma 3.3) to get

$$
\Pr\left[y^2 > \frac{10\log(10/\delta)}{cn_2}\right] \leq \delta/4.
$$

Thus, $d_{\epsilon/2}\left(\mathcal{M}(\mathbf{X}) \,\|\, \mathcal{M}(\tilde{\mathbf{X}})\right) \leq \delta/4 + \delta/4 = \delta/2$.

In both cases, we have $d_{\epsilon/2}\left(\mathcal{M}(\mathbf{X}) \,\|\, \mathcal{M}(\tilde{\mathbf{X}})\right) \leq \delta/2$ under (3). This concludes our privacy proof.

**Accuracy Analysis.** Let $D = \mathsf{N}(\mu, \Sigma)$ for some unknown $\mu, \Sigma$. Consider the algorithm $\mathcal{A}'$ where there is no truncation and no halting. Since $I \preceq \Sigma \preceq \kappa \cdot I$, standard concentration bounds (Lemmas 3.2 and 3.4) imply that the truncation and halting are not applied in $\mathcal{A}$ with probability at least $1 - \alpha/2$. Therefore, we have $d_{\mathrm{tv}}(Q_{\mathcal{A},D}, Q_{\mathcal{A}',D}) \leq \alpha/2$.

Now, notice that the algorithm $\mathcal{A}'$ just outputs $Y := \frac{1}{n_1}\left(\sum_{i \in [n_1]} X_i\right) + \sqrt{1 - \frac{1}{n_1}} \cdot \left(\sum_{i \in [n_2]} a_i U_i\right)$; this output $Y \sim \mathsf{N}(\mu, \Sigma)$. In other words, we have $Q_{\mathcal{A}',D} = \mathsf{N}(\mu, \Sigma)$. Combining these, we can conclude that $\mathcal{A}$ is $\alpha$-accurate.

## C   A Simpler but Worse Algorithm for Unknown Bounded Covariance

In this section, we give a simpler algorithm for the unknown bounded covariance setting. Its sample complexity bound of $\tilde{O}\left(\frac{d^{3/2}}{\alpha\epsilon^2}\right)$ is worse than the one in Algorithm 2 and Theorem 4.3, but is still better than $\Omega\left(\frac{d^2}{\alpha^2} + \frac{d^2}{\alpha\epsilon}\right)$ for DP learning [27] for constant $\epsilon > 0$.

The simpler sampler is given in Algorithm 3 and its guarantee is given in the theorem below. Note that this algorithm is once again just a form of (scaled) Gaussian mechanism, whereas the algorithm in the main body (Algorithm 2) adds noise that is input dependent.

**Theorem C.1.** *Assume that $I \preceq \Sigma \preceq \kappa \cdot I$ for some $\kappa > 0$ and that $\|\mu\| \leq R$, there is an $\alpha$-accurate $(\epsilon, \delta)$-DP sampler with sample complexity*

$$n = O\left(\left(R^2 + \kappa^2 d \left(\log\left(\frac{d}{\alpha\epsilon}\right) + \log\log(1/\delta)\right)\right) \cdot \frac{d^{1/2}}{\alpha} \cdot \frac{\log(1/\delta)}{\epsilon^2}\right).$$

---

**Algorithm 3** BOUNDEDCOVARIANCEGAUSSIANSAMPLER

---

**Parameters:** $B, \sigma > 0$, and $n_1, n_2 \in \mathbb{N}$.
Sample $X_1, \ldots, X_{n_1}, X_{n_1+1}, \ldots, X_{n_1+2n_2} \sim D$
**for** $i = 1, \ldots, n_1 + 2n_2$ **do**
$\quad\mid\quad X_i = \mathrm{trunc}_B^2(X_i)$
Sample $Z \sim \mathsf{N}(0, \sigma^2 I)$
**return** $\frac{1}{n_1}\left(\sum_{i \in [n_1]} X_i\right) + \sqrt{\frac{1-1/n_1}{2n_2}}\left(\sum_{i \in [n_2]}(X_{n_1+2i-1} - X_{n_1+2i})\right) + Z$

---

*Proof Sketch of Theorem C.1.* Let $B = R + 10^4 \kappa \sqrt{d\left(\log\left(\frac{2d}{\alpha\epsilon}\right) + \log\log(2/\delta)\right)}, \sigma = \frac{\sqrt{\alpha}}{2d^{1/4}}$ and

$n_1, n_2$ be such that $n_1 = n_2 \geq \frac{100B^2 \log(2/\delta)}{\sigma^2 \epsilon^2}$. Let $\mathcal{A}$ denote Algorithm 3 with parameters $B, n, \sigma$ as specified.

**Privacy Analysis.** $\mathcal{A}$ is the Gaussian mechanism with noise multiplier $\frac{\sigma}{B \cdot \sqrt{\frac{1-1/n_1}{2n_2}}} \geq 10\sqrt{\log(2/\delta)}/\epsilon$

by the setting of our parameters; therefore, $\mathcal{A}$ is $(\epsilon, \delta)$-DP.

**Accuracy Analysis.** To understand the accuracy guarantee, let $D = \mathsf{N}(\mu, \Sigma)$ for some unknown $\mu, \Sigma$ where $I \preceq \Sigma \preceq \kappa \cdot I$. Let us consider the algorithm $\mathcal{A}'$ where there is no truncation. Again, a standard concentration bound implies that the truncation is not applied anyway in $\mathcal{A}$ with probability at least $1 - \alpha/2$. Therefore, we have

$$d_{\mathrm{tv}}(Q_{\mathcal{A}, P}, Q_{\mathcal{A}', P}) \leq \alpha/2.$$

Now, notice that in algorithm $\mathcal{A}'$, we just output

$$Y := \frac{1}{n_1}\left(\sum_{i \in [n_1]} X_i\right) + \sqrt{\frac{1 - 1/n_1}{2n_2}}\left(\sum_{i \in [n_2]}(X_{n_1+2i-1} - X_{n_1+2i})\right) + Z.$$

Without truncation, this output $Y$ has identical distribution as $\mathsf{N}(\mu, \Sigma + \sigma^2 I)$. In other words, we have

$$d_{\mathrm{tv}}(Q_{\mathcal{A}', P}, \mathsf{N}(\mu, \Sigma)) = d_{\mathrm{tv}}(\mathsf{N}(\mu, \Sigma + \sigma^2 I), \mathsf{N}(\mu, \Sigma)) \leq \|\sigma^2 I\|_\Sigma \leq \|\sigma^2 I\|_F = \sigma^2 \cdot \sqrt{d} \leq \alpha/2,$$

where the first inequality follows from Lemma A.3 and the last inequality follows from our parameter selection. Combining the above two inequalities, we can conclude that the sampler is $\alpha$-accurate as desired. $\qquad\square$

## D   Gaussian Distributions: Lower Bounds

### D.1   Known Covariance

In this section, we prove a lower bound of $\Omega(\sqrt{d}/\epsilon)$ that holds even for the simplest case of known covariance (Theorem 1.3), showing that the dependence on $d$ in our sampler (Theorem 4.2) is nearly optimal.

We prove this by using a DP sampler to draw a large, but constant, number of samples and the use them to perform mean estimation; the lower bound for mean estimation (Theorem A.6) then gives the desired lower bound for the sample complexity of DP sampler.

*Proof of Theorem 1.3.* Given an $\alpha$-accurate $(\epsilon, \delta)$-DP sampler $\mathcal{A}$, we construct an algorithm $\mathcal{M}$ for mean estimation as follows:

- For $i = 1, \ldots, 10^6$:
  - Run $\mathcal{A}$ on $n$ fresh samples from $D$ to get $Y_i$.
- Output $\hat{\mu} \in \mathbb{R}^d$ with $\hat{\mu}[j] := \text{clip}_{-1,1}(\text{median}(Y_1[j], \ldots, Y_{10^6}[j]))$

Since $\mathcal{A}$ is $\alpha$-accurate, each $Y_i$ is sampled from a distribution $D'$ that is $\alpha$-close (in total variation distance) to $D = \mathsf{N}(\mu, I)$. For $\alpha = 0.1$, this means that

$$\Pr[Y_i(j) \leq \mu - 0.3] \leq \Phi(-0.3) + \alpha \leq 0.49,$$

and similarly,

$$\Pr[Y_i(j) \geq \mu + 0.3] \leq (1 - \Phi(0.3)) + \alpha \leq 0.49.$$

As a result, standard concentration bounds (e.g., Theorem E.1) imply that

$$\Pr[\text{median}(Y_1(j), \ldots, Y_{10^6}(j)) \in [\mu - 0.3, \mu + 0.3]] > 0.99.$$

This in turn implies that

$$\mathbb{E}_{\mathbf{X} \sim D^N, \hat{\mu} \sim \mathcal{M}(\mathbf{X})}[\|\hat{\mu} - \mu\|_2^2] \leq d \cdot \left(0.01 \cdot 2^2 + 0.99 \cdot 0.3^2\right) \leq d/6.$$

Applying Theorem A.6 with $\gamma = \sqrt{d/6}$, we can conclude that the sample complexity of $\mathcal{M}$ (which is equal to $10^6 n$) must be at least $\Omega\left(\frac{d}{\epsilon\sqrt{d/6}}\right) = \Omega(\sqrt{d}/\epsilon)$. Thus, we must have $n \geq \Omega(\sqrt{d}/\epsilon)$ as claimed. $\qquad\square$

## D.2 Unknown Bounded Covariance

Next, we will prove the lower bound for the unknown bounded covariance case (Theorem 1.5).

### D.2.1 Reduction to Covariance Estimation

---
**Algorithm 4** COVARIANCEESTIMATOR

---
**Parameters:** $N, n, L \in \mathbb{N}$, sampler $\mathcal{A}_{\text{sampler}}$, agnostic learner $\mathcal{A}_{\text{learner}}$ for centered Gaussians
Sample $X_1, \ldots, X_N \sim P$
**for** $\ell = 1, \ldots, L$ **do**
$\quad$ Randomly draw $n$ subsamples $X_{i_1^\ell}, \ldots, X_{i_n^\ell}$ without replacement from $X_1, \ldots, X_N$
$\quad$ $Y_\ell \leftarrow \mathcal{A}_{\text{sampler}}(X_{i_1^\ell}, \ldots, X_{i_n^\ell})$
$\mathsf{N}(0, \hat{\Sigma}) \leftarrow \mathcal{A}_{\text{learner}}(Y_1, \ldots, Y_L)$
**if** $0.5I \preceq \hat{\Sigma} \preceq 2.5I$ **then**
$\quad$ $\hat{\Sigma}' \leftarrow \hat{\Sigma}$
**else**
$\quad$ $\hat{\Sigma}' \leftarrow I$
**return** $\hat{\Sigma}'$

---

As stated earlier in Section 2, we will prove this lower bound by reducing to covariance estimation (Theorem A.7). This is done by first taking $N \gg n$ samples, and then generating each $Y_\ell$ by subsampling the input to $n$ samples and running our DP sampler. These $Y_\ell$'s are then feed into the agnostic learner to produce an estimate $\hat{\Sigma}$ for $\Sigma$. The full reduction is given in Algorithm 4. Note that here $\mathcal{A}_{\text{sampler}}$ will be the $(\epsilon, \delta)$-DP sampler and $\mathcal{A}_{\text{learner}}$ will be the learner from Theorem A.8.

**Privacy Analysis.** The privacy guarantee of Algorithm 4 follows easily from the amplification by subsampling and advanced composition of DP. This is formalized below.

**Lemma D.1.** *For any* $\epsilon^* \in (0, 1], \delta^* \in (0, 1)$, *if* $\mathcal{A}_{\text{sampler}}$ *is* $(\epsilon, \delta)$-*DP for* $\epsilon \leq \min\left\{1, \frac{\epsilon^* N}{4n\sqrt{L \ln(2/\delta^*)}}\right\}, \delta \leq \left(\frac{0.5N}{Ln}\right)\delta^*$, *then Algorithm 4 is* $(\epsilon^*, \delta^*)$-*DP.*

*Proof.* First, we apply Theorem A.4 which means that computing a single $Y_\ell$ is $(\epsilon', \delta')$-DP where

$$\epsilon' = \ln(1 + (n/N)(e^\epsilon - 1)), \qquad\qquad \delta' = (n/N) \cdot \delta.$$

Then, applying Theorem A.5 with $\delta_1 = 0.5\delta^*$ implies that all $(Y_1, \ldots, Y_L)$ together is $(\epsilon_1, L\delta' + \delta_1)$-DP where

$$\epsilon_1 = \left( \sqrt{2L \ln(1/\delta_1)} + L(e^{\epsilon'} - 1) \right) \epsilon'.$$

Combining the above expressions, we have

$$\epsilon_1 = \left( \sqrt{2L \ln(2/\delta^*)} + L(n/N)(e^\epsilon - 1) \right) \ln(1 + (n/N)(e^\epsilon - 1))$$

$$(\text{from } \epsilon \leq 1) \leq \left( \sqrt{2L \ln(2/\delta^*)} + L(n/N)(2\epsilon) \right) \ln(1 + (n/N)(2\epsilon))$$

$$\leq \left( \sqrt{2L \ln(2/\delta^*)} + L(n/N)(2\epsilon) \right) \cdot ((n/N)(2\epsilon))$$

$$\left( \text{from } \epsilon \leq \frac{\epsilon^* N}{4n\sqrt{L \ln(2/\delta^*)}} \right) \leq \epsilon^*,$$

and $L\delta' + \delta_1 \leq \delta^*/2 + \delta^*/2 \leq \delta^*$. In other words, $(Y_1, \ldots, Y_L)$ together is $(\epsilon^*, \delta^*)$-DP. Finally, applying $\mathcal{A}_{\text{learner}}$ on $(Y_1, \ldots, Y_L)$ is simply a post-processing step. Thus, the entire algorithm is $(\epsilon^*, \delta^*)$-DP as desired. $\qquad \square$

### D.2.2   Accuracy Analysis

Next, we argue that the accuracy of the covariance estimation algorithm, assuming the accuracy of the sampler and the agnostic learner.

**Lemma D.2.** *Let $\xi \in (0, 0.01]$ and $n, N, L \in \mathbb{N}$ be such that $N \geq \left( \frac{10nLd}{\xi} \right)^2$, and suppose that $I \preceq \Sigma \preceq 2I$. Furthermore, suppose that*

- *$\mathcal{A}_{\text{sampler}}$ is an $\left( \frac{\xi}{10C} \right)$-accurate sampler for the class of Gaussians under the assumption $I \preceq \Sigma \preceq 2I$, and*
- *$\mathcal{A}_{\text{learner}}$ is an $\left( \frac{\xi}{10C}, \frac{\xi^2}{200d^2} \right)$-accurate agnostic learner for the class of centered Gaussians,*

*where $C$ is the constant in Lemma A.2. Then, $\mathbb{E}[\|\hat{\Sigma}' - \Sigma\|_\Sigma^2] \leq \xi^2$ where $\hat{\Sigma}'$ denotes the output of Algorithm 3.*

*Proof.* Let $D = \mathsf{N}(0, \Sigma)$ denote the underlying distribution and for notational convenience, let $\zeta := \left( \frac{\xi}{10C} \right)$. Let $\mathcal{E}_{\text{disjoint}}$ denote the event that $i_1^1, \ldots, i_n^L$ are all different. Note that we have

$$\Pr[\neg \mathcal{E}_{\text{disjoint}}] \leq \sum_{\ell \neq \ell' \in [L], j, j' \in [n]} \Pr[i_j^\ell = i_{j'}^{\ell'}] = \sum_{\ell \neq \ell' \in [L], j, j' \in [n]} \frac{1}{N^2} \leq \frac{L^2 n^2}{2N} \leq \frac{\xi^2}{200d^2},$$

where the last inequality follows from our choice of parameters.

Next, suppose that $\mathcal{E}_{\text{disjoint}}$ holds. Then, we have that $Y_1, \ldots, Y_L$ are independently sampled from $Q_{\mathcal{A}_{\text{sampler}}, D}$. Applying the agnostic learning guarantee of $\mathcal{A}_{\text{learner}}$, with probability $1 - \frac{\xi^2}{200d^2}$, we get

$$d_{\text{tv}}(\mathsf{N}(0, \hat{\Sigma}), Q_{\mathcal{A}_{\text{sampler}}, D}) \leq 3 \cdot d_{\text{tv}}(\mathsf{N}(0, \Sigma), Q_{\mathcal{A}_{\text{sampler}}, D}) + \zeta.$$

Recall also from the accuracy guarantee of the sampler that $d_{\text{tv}}(\mathsf{N}(0, \Sigma), Q_{\mathcal{A}_{\text{sampler}}, D}) \leq \zeta$.

Combining all of these together we have

$$\Pr[d_{\text{tv}}(\mathsf{N}(0, \hat{\Sigma}), Q_{\mathcal{A}_{\text{sampler}}, D}) > 4\zeta] \leq 1 - \frac{\xi^2}{100d^2}.$$

Applying Lemma A.2, we get

$$\Pr[\|\hat{\Sigma} - \Sigma\|_\Sigma > \xi/2] \leq 1 - \frac{\xi^2}{100d^2}.$$

Notice that if $\|\hat{\Sigma} - \Sigma\|_\Sigma \leq \xi/2$, then we have $\hat{\Sigma}' = \hat{\Sigma}$. Furthermore, $I \preceq \Sigma \preceq 2I$ and $0.5I \preceq \hat{\Sigma}' \preceq 2.5I$ imply that $\|\hat{\Sigma}' - \Sigma\|_\Sigma \leq 6d$. Thus, we have

$$\mathbb{E}[\|\hat{\Sigma}' - \Sigma\|_\Sigma^2] \leq (\xi/2)^2 + (6d)^2 \cdot \Pr[\|\hat{\Sigma} - \Sigma\|_\Sigma > \xi/2] \leq (\xi/2)^2 + (6d)^2 \cdot \frac{\xi^2}{100d^2} \leq \xi^2. \quad \square$$

### D.2.3 Putting Things Together

Combing the privacy and accuracy guarantees and plugging in the appropriate parameters immediately yields Theorem 1.5.

*Proof of Theorem 1.5.* Let $\alpha = \xi/(10C)$ where $\xi$ is as in Theorem A.7 and $C$ is as in Lemma A.2. Suppose for the sake of contradiction that there exists an $(\epsilon, \delta)$-DP $\alpha$-accurate sampler with sample complexity $n = o\left(\frac{d}{\epsilon\sqrt{\log d}}\right)$ under the assumption $I \preceq \Sigma \preceq 2I$. Then, let us select the parameters as follows:

- $L \le O\left(d^2\right)$ denote the sample complexity of the $\left(\frac{\xi}{10C}, \frac{\xi^2}{200d^2}\right)$-accurate agnostic learner for the class of centered Gaussians as given by Theorem A.8.
- $N = \left\lceil \left(\frac{10nLd}{\xi}\right)^2 \right\rceil \le O\left(n^2 d^2 \log d\right)$.
- $\delta^* = O\left(\min\left\{1/N, d^2/(N \log N)\right\}\right)$. Note that, when the constant in big-O notation is sufficiently large, our choice of parameters implies that $\delta \le \left(\frac{0.5N}{Ln}\right)\delta^*$.
- $\epsilon^* = \frac{4n\epsilon\sqrt{L\ln(2/\delta^*)}}{N} = o(d^2/N)$.

By Lemma D.2, we have that $\mathbb{E}[\|\hat{\Sigma}' - \Sigma\|_{\hat{\Sigma}}^2] \le \xi^2$. Furthermore, by Lemma D.1, we have that the algorithm COVARIANCEESTIMATOR is $(\epsilon^*, \delta^*)$-DP. However, we also have $N = o(d^2/\epsilon^*)$, which contradicts Theorem A.7. $\qquad\square$

# E  Product Distributions on $\{0,1\}^d$

In this section, we describe and analyze our sampler for product distributions on $\{0,1\}^d$ (Theorem 1.7). We may assume w.l.o.g. that $p_1, \ldots, p_d \le 3/4$. (Otherwise, we first privately estimate $p_i$ using, e.g., the Laplace mechanism, and flip the $i$th bit in all samples if the estimate is more than $3/4$.)

Given a dataset element $X_i \in \{0,1\}^d$, we use $X_i[j]$ to denote its $j$th coordinate, and $X_i[S] = (X_i[j])_{j \in S}$ to denote the vector $X_i$ restricted to the subset $S \subseteq [d]$ of coordinates.

## E.1  Additional Preliminaries

Here we list a few concentration inequalities that are useful. We start with the following version of the Bernstein inequality [see, e.g., 42, Theorem 2.8.4]:

**Theorem E.1** (Bernstein's inequality). *Let $Y_1, \ldots, Y_n$ be independent real-valued random variables such that, with probability 1, $Y_i \in [0, C]$ for all $i \in [n]$. Let $V = \sum_{i \in [n]} \mathrm{var}(Y_i)$. Then, for any $\Delta \ge 0$, we have*

$$\Pr\left[\left|\sum_{i \in [n]} Y_i - \sum_{i \in [n]} \mathbb{E}[Y_i]\right| > \Delta\right] \le 2\exp\left(\frac{-\Delta^2}{2V + C\Delta}\right).$$

We now list a couple of versions of this inequality, which will be more useful in our setting.

**Corollary E.2.** *Let $P$ be a product distribution on $\{0,1\}^d$. For any $\beta, \gamma \in (0, \frac{1}{2})$, let $n > \frac{50}{\gamma}\log\frac{d}{\beta}$ and $X_1, \ldots, X_n \sim P$. Let $\overline{X} := \frac{1}{n}\sum_{i \in [n]} X_i$. Then, we have*

$$\Pr\left[\forall j \in [d], \overline{X}[j] \in [0.9p_j - \gamma, 1.1p_j + \gamma]\right] \ge 1 - \beta.$$

*Proof.* For each $j \in [d]$, applying Theorem E.1 with $\Delta = \max\{\gamma, 0.1p_j\} \cdot n$ and using $p_j \le 3/4$, we obtain

$$\Pr\left[|\overline{X}[j] - p_j| \le \max\{\gamma, 0.1p_j\}\right] \le 2\exp\left(\frac{-\Delta^2}{2np_j + \Delta}\right) \le 2\exp\left(-\frac{\Delta}{21}\right) \le \frac{\beta}{d}.$$

Taking a union bound over all $j \in [d]$ completes the proof. $\qquad\square$

**Corollary E.3.** *Let $P$ be a product distribution on $\{0,1\}^d$. For any $\beta, \gamma \in (0, \frac{1}{2})$, let $n > \frac{50}{\gamma} \log \frac{d}{\beta}$ and $X_1, \ldots, X_n \sim P$. Then, we have*

$$\Pr\left[\forall i \in [n], \sum_{j \in [d]} \frac{X_i[j]}{\max\{p_j, \gamma\}} \leq 3d + \frac{4}{\gamma} \log \frac{n}{\beta}\right] \geq 1 - \beta.$$

*Proof.* Denote $Y_j = X_i[j]/\max\{p_j, \gamma\}$ for all $j \in [d]$. Here we have $\mathbb{E}[Y_j] = p_j/\max\{p_j, \gamma\} \leq 1$ and $\text{var}(Y_j) \leq p_j/\max\{p_j, \gamma\}^2 \leq 1/\gamma$. Moreover, with probability 1, we have $Y_j \leq 1/\gamma$. As a result, we can apply Theorem E.1 with $\Delta = 2d + 4\log(n/\beta)/\gamma$, which gives

$$\Pr\left[\left|\sum_{j \in [d]} \frac{X_i[j]}{\max\{p_j, \gamma\}} - d\right| > \Delta\right] \leq 2\exp\left(\frac{-t^2}{2d/\gamma + t/\gamma}\right) \leq 2\exp(-0.5\Delta\gamma) \leq \frac{\beta}{n}.$$

Taking a union bound over all $i \in [n]$ completes the proof. $\qquad\square$

### E.2 Private Preconditioner

We start with the following private preconditioner, which estimates each $p_j$ up to a constant factor. This preconditioner is similar to that of [27], except that we add noise from the Laplace distribution[13] (instead of Gaussian noise) to achieve pure-DP and that we are looking for a coarser guarantee compared to [27].

**Lemma E.4.** *There exists an $\epsilon$-DP algorithm that takes*

$$O\left(\left(d\log\left(\frac{d}{\alpha}\right) + \frac{d}{\alpha}\right) \frac{\log\left(\frac{d}{\alpha\beta}\right)}{\epsilon}\right),$$

*samples as input and output integers $\ell_1, \ldots, \ell_d \in \{0, \ldots, \lceil \log(\frac{2d}{\alpha}) \rceil\}$ such that, with probability $1 - \beta$, the following hold for all $j \in [d]$:*

- *if $\ell_j \neq \lceil \log(\frac{2d}{\alpha}) \rceil$, then $p_j \in [1/4 \cdot 2^{-\ell_j}, 3/4 \cdot 2^{-\ell_j}]$, and*
- *if $\ell_j = \lceil \log(\frac{2d}{\alpha}) \rceil$, then $p_j \leq \alpha/2d$.*

In the remaining of this section, we will focus on the proof of Lemma E.4 and Algorithm 5.

**Sample Complexity.** From Algorithm 5, the total sample complexity can be bounded by

$$\sum_{\ell \in [L]} n_\ell = O\left(\sum_{\ell \in [L]} (d + 2^\ell) \frac{\log\left(\frac{d}{\alpha\beta}\right)}{\epsilon}\right) = O\left(\left(d\log\left(\frac{d}{\alpha}\right) + \frac{d}{\alpha}\right) \frac{\log\left(\frac{d}{\alpha\beta}\right)}{\epsilon}\right),$$

as desired.

**Privacy Analysis.** In each iteration $\ell \in [L]$, the samples $X_1^\ell, \ldots, X_{n_\ell}^\ell$, after truncation, are noised with a Laplace distribution; by the privacy guarantee of the Laplace mechanism, each iteration is thus $\epsilon$-DP. Furthermore, since each iteration uses a fresh set of samples, by the parallel composition property of DP, the entire algorithm is $\epsilon$-DP as desired.

**Accuracy Analysis.** We will prove this by induction. Specifically, let $S_\ell^\uparrow := \{j \in [d] \mid p_j \leq 3/4 \cdot 2^{-\ell}\}$ and $S_\ell^\downarrow := \{j \in [d] \mid p_j \leq 1/4 \cdot 2^{-\ell}\}$. We will show that

$$\Pr\left[\forall \ell \in \{0, \ldots, t\}, S_{\ell-1}^\downarrow \subseteq S_\ell \subseteq S_\ell^\uparrow\right] \geq 1 - \frac{t\beta}{L}, \tag{6}$$

for all $t \in [L]$ by an induction on $t$. Observe that this statement implies the desired accuracy claim in the lemma because $j$ gets assigned $\ell_j = \ell$ if and only if $j \in S_\ell \setminus S_{\ell+1}$. (Here we use the convention that $S_{L+1} = \emptyset$.)

---

[13]The *Laplace distribution* $\mathsf{Lap}(b)$ is given by the PDF $f_{\mathsf{Lap}(b)}(x) \propto \exp(-|x|/b)$.

**Algorithm 5** PRECONDITIONERPRODUCTDIST

---

**Input:** An unknown product distribution $P$ over $\{0,1\}^d$, privacy parameter $\epsilon$, accuracy parameter $\alpha$, failure probability parameter $\beta$

**Output:** $\ell_1, \ldots, \ell_d \in \{0, 1, \ldots, \lceil \log(\frac{2d}{\alpha}) \rceil\}$

$L \leftarrow \lceil \log(\frac{2d}{\alpha}) \rceil$
$\ell_j \leftarrow 0$ for every $j \in [d]$
$S_0 \leftarrow [d]$
**foreach** $\ell = 0, 1, \ldots, L-1$ **do**

$\quad B_\ell \leftarrow 1000(d \cdot 2^{-\ell} + 1)$

$\quad n_\ell \leftarrow \frac{1000}{\epsilon} \cdot B_\ell \cdot 2^\ell \cdot \log\left(\frac{d}{\alpha\beta}\right)$

$\quad$ Sample $X_1^\ell, \ldots, X_{n_\ell}^\ell \sim P$

$\quad S_{\ell+1} \leftarrow \emptyset$

$\quad q_\ell[S_\ell] \leftarrow \dfrac{1}{n_\ell} \left( \mathsf{Lap}\left(\dfrac{B_\ell}{\epsilon}\right) + \sum_{i \in [n_\ell]} \mathrm{trunc}_{B_\ell}^1(X_i^\ell[S_\ell]) \right)$

$\quad$ **foreach** $j \in S_\ell$ **do**

$\quad\quad$ **if** $q_\ell[j] \leq 0.57 \cdot 2^{-\ell}$ **then**

$\quad\quad\quad$ Add $j$ to $S_{\ell+1}$

$\quad\quad$ **else**

$\quad\quad\quad \ell_j \leftarrow \ell$

**foreach** $j \in S_L$ **do**

$\quad \ell_j \leftarrow L$

**return** $\ell_1, \ldots, \ell_d$

---

**Base Case.** (2) trivially holds for $t = 0$ as $S_0^\uparrow = S_0 = [d]$.

**Inductive Step.** Suppose that (6) holds for $t-1$ for some $t \in \mathbb{N}$. We will also show that it also holds for $t$. From a union bound, it suffices to show

$$\Pr\left[S_{t-1}^\downarrow \subseteq S_t \subseteq S_t^\uparrow \mid S_{t-1} \subseteq S_{t-1}^\uparrow\right] \geq 1 - \frac{\beta}{L}.$$

To show this, we need the following claim:

*Claim* 1. Assume $S_{t-1} \subseteq S_{t-1}^\uparrow$. For each $j \in S_t$, with probability $1 - \frac{0.5\beta}{Ld}$, we have

$$\frac{1}{n_t}\left(\sum_{i \in [n_t]} \mathrm{trunc}_{B_t}^1(X_i^t[S_t])\right)_j \in \left[0.8p_j - 0.01 \cdot 2^{-t}, 1.1p_j + 0.01 \cdot 2^{-t}\right].$$

Before we prove Claim 1, let us see how to use it to finish the proof. Let $Z^t \sim \mathsf{Lap}\left(B_t/\epsilon\right)$ be the Laplace noise added. First, by a standard concentration bound for the Laplace distribution, we have

$$\Pr\left[|Z_j^t| < \frac{B_t}{\epsilon}\log\left(\frac{4Ld}{\beta}\right)\right] \geq 1 - \frac{0.5\beta}{Ld}.$$

Note also that by our choice of parameters, we have $B_t/\epsilon \cdot \log\left(4Ld/\beta\right) \leq 0.01 \cdot 2^{-t} \cdot n_t$.

Hence, by a union bound, we have that with probability $1 - \beta/L$ for all $j \in S_t$,

$$q_t[j] \in [0.8p_j - 0.02 \cdot 2^{-t}, 1.1p_j + 0.02 \cdot 2^{-t}].$$

Consider any $j \in S_t \setminus S_t^\uparrow$. Since $p_j \geq 0.75 \cdot 2^{-t}$, we have $q_t[j] \geq 0.58 \cdot 2^{-t} > \tau_t$. Hence, $j$ will not be included in $S_t$, coming to a contradiction. In other words, $S_t \setminus S_t^\uparrow = \emptyset$, i.e., $S_t \subseteq S_t^\uparrow$. Similarly, consider any $j \in S_{t-1}^\downarrow$. Since $p_j \leq 0.25 \cdot 2^{-t+1}$, we have $q_t[j] \leq 0.57 \cdot 2^{-t} = \tau_t$. By Algorithm 5, $j$ will be included in $S_t$. Hence, $S_{t-1}^\downarrow \subseteq S_t$. These conclude the inductive step.

We are now left to prove Claim 1.

*Proof of Claim 1.* Let $Y_i = \left(\mathrm{trunc}_{B_t}^1(X_i^t[S_t])\right)_j$.

First note that $\sum_{i \in [n_t]} Y_i \leq \sum_{i \in [n_t]} X_i^t[j]$. Therefore, we may apply Corollary E.2, which gives

$$\Pr\left[\frac{1}{n_t} \cdot \sum_{i \in [n_t]} Y_i \leq 1.1 p_j + 0.01 \cdot 2^{-t}\right] > 1 - \frac{0.25\beta}{Ld}.$$

Next, to give a lower bound on $Y_i$, we also observe that

$$\begin{aligned}
\Pr[Y_i = 1] &= \Pr\left[X_i^t[j] = 1 \text{ and } \|X_i^\ell[S_t]\|_1 \leq B_t\right] \\
&\geq \Pr[X_i^t[j] = 1 \text{ and } \|X_i^t[S_t \setminus \{j\}]\|_1 \leq B_t - 1] \\
&= \Pr[X_i^t[j] = 1] \cdot \Pr[\|X_i^t[S_t \setminus \{j\}]\|_1 \leq B_t - 1] \\
&= p_j \cdot \Pr[\|X_i^t[S_t \setminus \{j\}]\|_1 \leq B_t - 1],
\end{aligned}$$

and

$$\mathbb{E}[\|X_i^t[S_t \setminus \{j\}]\|_1] = \sum_{j' \in S_t \setminus \{j\}} p_{j'}.$$

We note that $S_t \subseteq S_{t-1}$ holds by Algorithm 5 and $S_{t-1} \subseteq S_{t-1}^\uparrow$ follows the assumption. Hence, $p_{j'} < 0.75 \cdot 2^{-(t-1)}$ for every $j' \in S_{t-1}^\uparrow$. Moreover, $|S_{t-1}^\uparrow| \leq d$. Putting everything together, we can further bound

$$\mathbb{E}[\|X_i^t[S_t \setminus \{j\}]\|_1] = \sum_{j' \in S_t \setminus \{j\}} p_{j'} \leq \sum_{j' \in S_{t-1}^\uparrow} p_{j'} \leq d \cdot 0.75 \cdot 2^{-(t-1)} \leq 0.1(B_t - 1).$$

Therefore, by Markov's inequality, we have

$$\Pr[\|X_i^t[S_t \setminus \{j\}]\|_1 \leq B_t - 1] \geq 0.9.$$

Combining with the above, we have $\Pr[Y_i = 1] \geq 0.9 p_j$. Notice also that $Y_1, \ldots, Y_{n_t}$ are i.i.d. and always lie between $[0, 1]$. Thus, we can apply Theorem E.1 with $\Delta = (0.1 p_j + 0.01 \cdot 2^{-t}) n_t$ to obtain

$$\Pr\left[\frac{1}{n_t} \cdot \sum_{i \in [n_t]} Y_i < 0.8 p_j - 0.01 \cdot 2^{-t}\right] \leq 2 \exp\left(\frac{\Delta^2}{2 p_j n_t + \Delta}\right) \leq 2 \exp\left(\frac{\Delta^2}{21\Delta}\right) \leq \frac{0.25\beta}{Ld}.$$

Applying a union bound then yields the claim. $\qquad\square$

### E.3  Sampler

Next, we give a DP sampler for an unknown product distribution, assuming that a rough estimate of each $p_j$ is already computed via the private preconditioner in Section E.2. The exact guarantee is given below; combining this with Lemma E.4, we immediately arrive at Theorem 1.7.

**Lemma E.5.** *Suppose that $\ell_1, \ldots, \ell_d$ satisfy the conditions in Lemma E.4. Then, there is an $\epsilon$-DP algorithm taking $\ell_1, \ldots, \ell_d$ as input, that is an $\alpha$-accurate sampler and has a sample complexity of*

$$O\left(\frac{d \log(d/\alpha)}{\alpha\epsilon}\right).$$

**Algorithm 6** PRODUCTDISTSAMPLER

---

**Input:** An unknown product distribution $P$ over $\{0,1\}^d$, privacy parameter $\epsilon$, accuracy parameter $\alpha$, and $\ell_1, \ldots, \ell_d \in \mathbb{N}$ satisfy the conditions in Lemma E.4
**Output:** A random sample that is $\alpha$-accurate for $P$
$B \leftarrow 1000(d/\alpha) \cdot \log(d/\alpha)$
$n \leftarrow \lceil 16B/\epsilon \rceil$
Sample $X_1, \ldots, X_n \sim P$
$w \leftarrow (2^{\ell_1}, \ldots, 2^{\ell_d})$
$q \leftarrow \frac{1}{n} \cdot \sum_{i \in [n]} \text{trunc}^1_{B,w}(X_i)$
**for** $j = 1, \ldots, d$ **do**
$\quad \tilde{p}_j \leftarrow \text{clip}_{\frac{1}{8w_j}, \frac{7}{8w_j}}(q_j)$
Sample $Z \sim \text{Ber}(\tilde{p}_1) \otimes \cdots \otimes \text{Ber}(\tilde{p}_d)$
**return** $Z$

---

In the following, we will focus on the proof of Lemma E.5 with Algorithm 6.

**Privacy Analysis.** Consider any pair $\mathbf{X}, \mathbf{X}'$ of neighboring datasets. We may assume w.l.o.g. that the two datasets agree except for $X_n$ and $X'_n$. We write $q', \tilde{p}'$ to denote the values of $q, \tilde{p}$ computed for $\mathbf{X}'$. Now, consider any possible output $y \in \{0,1\}^d$. For each $j \in [d]$, we claim that

$$\frac{\Pr[\mathcal{A}(\mathbf{X}')_j = y_j]}{\Pr[\mathcal{A}(\mathbf{X})_j = y_j]} \leq 1 + 8w_j |\tilde{p}'_j - \tilde{p}_j|. \tag{7}$$

To show this, we first mention two facts:

- $\dfrac{\Pr[\mathcal{A}(\mathbf{X}')_j = 0]}{\Pr[\mathcal{A}(\mathbf{X})_j = 0]} = \dfrac{1 - \tilde{p}'_j}{1 - \tilde{p}_j} \leq 1 + \dfrac{|\tilde{p}'_j - \tilde{p}_j|}{1 - \tilde{p}_j}$;
- $\dfrac{\Pr[\mathcal{A}(\mathbf{X}')_j = 1]}{\Pr[\mathcal{A}(\mathbf{X})_j = 1]} = \dfrac{\tilde{p}'_j}{\tilde{p}_j} \leq 1 + \dfrac{|\tilde{p}'_j - \tilde{p}_j|}{\tilde{p}_j}$.

It suffices to show $1/(8w_j) \leq \tilde{p}_j \leq 1 - 1/(8w_j)$ to prove (7). This follows since $w_j > 1$ and $1/(8w_j) \leq \tilde{p}_j \leq 7/(8w_j)$ due to clipping.

We also note the following useful property:

$$\|w \circ (q' - q)\|_1 = \frac{1}{n} \cdot \|w \circ (\text{trunc}^1_{B,w}(X'_n) - \text{trunc}^1_{B,w}(X_n))\|_1$$
$$\leq \frac{1}{n} \cdot \|w \circ (\text{trunc}^1_{B,w}(X'_n) + \text{trunc}^1_{B,w}(X_n))\|_1 \leq \frac{2B}{n}. \tag{8}$$

Finally, we are ready to prove the privacy guarantee:

$$\frac{\Pr[\mathcal{A}(\mathbf{X}') = y]}{\Pr[\mathcal{A}(\mathbf{X}) = y]} = \prod_{j \in [d]} \frac{\Pr[\mathcal{A}(\mathbf{X}')_j = y_j]}{\Pr[\mathcal{A}(\mathbf{X})_j = y_j]} \leq \prod_{j \in [d]} \left(1 + 8w_j \cdot |\tilde{p}'_j - \tilde{p}_j|\right)$$
$$\leq \prod_{j \in [d]} \exp\left(8w_j \cdot |\tilde{p}'_j - \tilde{p}_j|\right) = \exp\left(8w \circ |\tilde{p}' - \tilde{p}|\right)$$
$$\leq \exp\left(8w \circ |q' - q|\right) \leq \exp\left(\frac{16B}{n}\right) = \exp(\epsilon),$$

where the third inequality is due to clipping, the penultimate step follows (8), and the last step follows from our choice of parameters. Thus, the algorithm is $\epsilon$-DP as claimed.

**Accuracy Analysis.** Let $S = \{j \in [d] \mid p_j < 2^{-L}\}$. We first consider the version of $\mathcal{A}$ (Algorithm 6), where there is no truncation at all and there is no clipping for any $j \notin S$, denoted as $\mathcal{A}'$. Let $Q_{\mathcal{A}',P} = \text{Ber}(p_1^*) \otimes \cdots \otimes \text{Ber}(p_d^*)$. Fix $\beta = \alpha/12d < \alpha/4$. We define the event $\mathcal{E}$ as "no $X_i$ is truncated for any $i \in [n]$ and no $p_j$ is clipped for any $j \notin S$ by Algorithm 6". By concentration results (Corollaries E.2 and E.3 with $\gamma = 2^{-L-1}$) and our selection of parameters, we can conclude that $\mathcal{E}$ happens with probability at least $1 - \beta$, and therefore

$$d_{\text{tv}}(Q_{\mathcal{A}',P}, Q_{\mathcal{A},P}) = d_{\text{tv}}(Q_{\mathcal{A}',P}, Q_{\mathcal{A},P} \mid \mathcal{E}) \cdot \Pr[\mathcal{E}] + d_{\text{tv}}(Q_{\mathcal{A}',P}, Q_{\mathcal{A},P} \mid \bar{\mathcal{E}}) \cdot \Pr[\bar{\mathcal{E}}]$$

$$= 0 \cdot (1 - \beta) + 1 \cdot \beta \leq \alpha/2.$$

Furthermore, we can see that $\mathbb{E}[p_j^*] = p_j$ if $j \notin S$. For $j \in S$, clipping ensures us that $p_j^* \leq 2^{-L}$, hence $\mathbb{E}[p_j^*] \leq 2^{-L}$. Together, we have

$$d_{\mathrm{tv}}(Q_{\mathcal{A}',P}, P) = \sum_{j \in [d]} |\mathbb{E}[p_j^*] - p_j| = \sum_{j \in S} |\mathbb{E}[p_j^*] - p_j| \leq \sum_{j \in S} 2^{-L} \leq d \cdot 2^{-L} \leq \alpha/2.$$

Combining the above two inequalities yields the $\alpha$-accuracy guarantee of $\mathcal{A}$:

$$d_{\mathrm{tv}}(Q_{\mathcal{A},P}, P) \leq d_{\mathrm{tv}}(Q_{\mathcal{A},P}, Q_{\mathcal{A}',P}) + d_{\mathrm{tv}}(Q_{\mathcal{A}',P}, P) \leq \alpha/2 + \alpha/2 = \alpha.$$

*Proof of Theorem 1.7.* Finally, we are ready to prove Theorem 1.7. For simplicity, let $\alpha'$ be the target accuracy parameter. Set $\alpha = \alpha'/2$ and $\beta = \alpha/12d < \alpha'/2$. We define the event $\mathcal{E}$ as "Algorithm 5 returns $\ell_1, \ldots, \ell_d$ satisfying the properties in Lemma E.4". If $\mathcal{E}$ holds, then the error of our sampler is at most $\alpha'/2$, which is implied by Lemma E.5. If $\mathcal{E}$ fails, the error is at most 1, but this event happens with probability at most $\beta < \alpha'/2$. Together, the error of our sampler is

$$\alpha'/2 \cdot (1 - \beta) + 1 \cdot \beta < \alpha'/2 + \alpha'/2 = \alpha',$$

and the sampling complexity is (with $\beta = \alpha'/24d$)

$$O\left( \left( d \log\left(\frac{d}{\alpha'}\right) + \frac{d}{\alpha'} \right) \frac{\log\left(\frac{d}{\alpha'\beta}\right)}{\epsilon} + \frac{d}{\alpha'\epsilon} \log\left(\frac{d}{\alpha'}\right) \right) = O\left( \frac{d}{\epsilon} \log^2\left(\frac{d}{\alpha'}\right) + \frac{d}{\alpha'\epsilon} \log\left(\frac{d}{\alpha'}\right) \right),$$

as desired. □

