# OpenReview forum: "On Differentially Private Sampling from Gaussian and Product Distributions"
_NeurIPS.cc/2023/Conference — NeurIPS 2023 poster_

### Official Review · Reviewer_Tiqw · 2023-06-28

**Soundness:** 3 good
**Presentation:** 3 good
**Contribution:** 2 fair
**Rating:** 6
**Confidence:** 2

**Summary:**

The paper discusses the problem of generating a sample from a distribution that is close to an unknown distribution P, under the constraint of differential privacy. The authors present new differential privacy sampling algorithms for multi-dimensional Gaussian distributions, with different assumptions on the available information. They also obtain a pure-DP algorithm for product distributions on the binary hypercube. The paper aims to bridge the quantitative gap between DP distribution sampling and DP distribution learning.

**Strengths:**

The paper proposes a new DP sampling algorithm for multivariate Normal and the product Bernoulli distribution.
It also aims to bridge the gap between DP distribution sampling and DP distribution learning. It could lead to more effective and efficient methods for privacy-preserving machine learning.



**Weaknesses:**

The scope of the paper is quite limited as it only focuses on Gaussian and binary product distributions. While these are important classes of distributions, the applicability of the proposed methods to other types of distributions remains unclear. It would be beneficial if the authors could extend their analysis to a broader range of distributions or discuss the potential challenges and strategies for doing so.

The paper indicates that fewer samples are needed for estimation under these specific parameter families compared to sampling. It's not clear whether this is due to the geometry of the parameter space and output space, or if it's more related to the properties of differential privacy itself, perhaps due to the use of the composition theorem. It would be interesting to see a discussion on whether similar situations occur in non-DP settings. This could help clarify whether the observed phenomena are intrinsic to differential privacy or if they are more general statistical phenomena.

**Questions:**

See weakness

---

> ### Author Rebuttal · Authors · 2023-08-08
>
> * While it would be nice to have a more generic algorithm, we note that it is usually quite difficult to nail down the exact sample complexity in DP for a generic family of distributions. To give a sense of this difficulty, note that it required more than a dozen papers (cited in our paper) focusing almost solely on learning of Gaussian distributions to reach the current state-of-the-art results (which nonetheless are still not completely tight in some regime of parameters). This is the motivation for us to focus on Gaussian and product distributions in this work. We do agree that extending this to other distributions is a very interesting direction and indeed we had included this in the last section (Line 361).
>
> * Regarding the intuition, note that without privacy constraints, the sampling problem is trivial: just output the input sample. This requires only a single sample and has no error at all! Therefore, sampling is also easier than learning in the non-private setting as well. On the other hand, the privacy constraints make the sampling problem much more challenging which is why it is unclear whether it is even easier than learning—the question we answer in this paper for Gaussians.

---

> > ### Comment · Reviewer_Tiqw · 2023-08-16
> >
> > Thank you for your response, I have raised the score to 6, I don't have further questions.

---

### Official Review · Reviewer_iEFD · 2023-07-04

**Soundness:** 3 good
**Presentation:** 3 good
**Contribution:** 3 good
**Rating:** 7
**Confidence:** 2

**Summary:**

This work studies the problem of differentially private sampling. In this setting, the goal is the following: Given $n$ iid samples from an unknown distribution (e.g., a Gaussian distribution with an unknown mean), design a private algorithm that produces a fresh sample from the distribution the original samples came from. This framework was recently introduced with the following motivation: Sometimes it would be useful to have access to a synthetic dataset, i.e., a dataset that behaves in the same way as the original one but doesn't reveal any information about it.

The authors systematically investigate this problem for various cases of the gaussian distribution. They give algorithms and nearly matching lower bounds. They also have some results for the binary setting.

**Strengths:**

In my eyes, this paper makes important contributions to a new interesting line of work. In particular, it shows that, at least for the Gaussian distribution, the private sampling task is inherently easier than the associated learning task, e.g., learning the mean. For the former, the dependence on the accuracy parameters is only logarithmic, while for the latter it is polynomial.

The authors investigate this task in a systematic way, in the gaussian case, they study three different natural settings (known covariance, unknown and bounded covariance, and unknown and unbounded covariance) and give upper and lower bounds.

**Weaknesses:**

For the discrete setting, the dependence on the accuracy parameter is qualitatively worse (polynomial vs. logarithmic). It would be nice to give some intuition if this necessary and if so, why. For instance, is there any lower bound that you can prove? Even one that is not necessarily tight.

**Questions:**

Related to my comment above: Do you believe the discrete setting is inherently harder? Could it be that the difference between learning and sampling is less stark in this setting?

**Limitations:**

The authors discuss limitations and open questions related to their work.

---

> ### Author Rebuttal · Authors · 2023-08-08
>
> The polynomial dependence on $1/\\alpha$ is necessary in the product distribution case; as indicated in Table 2, it was already shown in [34] that the sample complexity for $(\\epsilon,\\delta)$-DP is at least $\\Omega(d/\\alpha\\epsilon)$. As you said, it is indeed a pretty interesting situation, since we can get a polylogarithmic dependency for Gaussians but (provably) not for product distributions. We will make this clearer in the revision.
>
> Since the dependency of $\\alpha$ in our algorithm is essentially tight in the product distribution case, we kindly ask the reviewer to adjust the score accordingly.

---

> > ### Comment · Reviewer_iEFD · 2023-08-14
> >
> > Thank you for clarifying this for me. Since this alleviated my main concern, I have indeed increased my score.

---

### Official Review · Reviewer_cbpB · 2023-07-06

**Soundness:** 3 good
**Presentation:** 3 good
**Contribution:** 2 fair
**Rating:** 5
**Confidence:** 4

**Summary:**

The paper considers the problem of sampling from distributions under differential privacy. In particular, the specific formulation presented in the paper assumes the existence of a dataset that has been drawn i.i.d. from a distribution $P$ that belongs in a family of distributions $\mathcal{F}$ and the goal is to a generate a sample from a distribution $Q \in \mathcal{F}$ such that $d_{\mathrm{TV}}(P, Q) \le \alpha$ in a way that protects the privacy of the input dataset. This problem is considered for various choices of $\mathcal{F}$, which include i) Gaussians with uknown mean but known covariance, ii) Gaussians with unknown mean and unknown covariance $\Sigma$ which satisfies $I \preceq \Sigma \preceq \kappa I$ for some known $\kappa$, iii) Gaussians with unknown mean and covariance without any a-priori bounds on $\Sigma$, iv) binary product distributions. The results include upper bounds for the aforementioned families, as well as lower bounds for families (i) and (ii).

**Strengths:**

The writing is fairly accessible and it is relatively easy to grasp the results and follow the exposition. In terms of contributions, I feel the main strength of the paper lies in the results about binary product distributions, which constitute significant improvements over prior work. Additionally, the arguments via which the lower bounds are established are quite elegant.

**Weaknesses:**

I have two main concerns with this work.

The first has to do with the bounds given for Gaussian distributions. In particular, while the guarantees given in the paper are tight (or nearly tight) with respect to the dimension $d$, it seems like the guarantees are sub-optimal in terms of the range parameters. For the case of Gaussians with known covariance but unknown mean, the dependence in the sample complexity for sampling under approximate differential privacy is polynomial with respect to $R$ (Theorem 4.2), where $R$ is a positive real number such that $|| \Sigma^{- \frac{1}{2}} \mu ||_2 \le R$. In [KV18], the authors show that, under approx-DP, you do not need any a-priori bounds on the mean to do parameter/density estimation for univariate Gaussians. This immediately extends to $d$-dimensional Gaussians with known covariance, since one can rescale the datapoints by $\Sigma^{- \frac{1}{2}}$ to make the Gaussian spherical and then apply the algorithm of [KV18] component-wise. Thus, it feels out of place for sampling to require a polynomial dependence on $R$, despite sampling being an easier task than estimation. Similarly, I have a similar concern for the guarantees of Theorem 4.3, both related to the dependence in $R$ and in $\kappa$.

The second concern (which is not as significant as the first one) has to do with the model considered. The standard formulation of the sampling problem involves having a known target distribution that you want to draw a sample from and defining a process that randomly generates points that come from a distribution that's close to the target (e.g., via a random walk). In this formulation, there is no input dataset to work with, thus not leaving much room for formulations that involve DP. In light of this, the variant of the standard sampling problem considered in this paper feels somewhat artificial. It might make sense in the context of e.g., synthetic data generation (a topic on which there is a broad literature), but it feels a bit odd to market this as ``sampling with DP". However, to be fair, this is a criticism that one could make about [RSSS21], which is the paper that introduced the model that the present paper considers. For that reason, this is not something I hold against this work and I consider it minor compared the points raised in the previous paragraph.

Overall, if the guarantees were to be optimized in terms of the range parameters, I believe a polished version of this paper could stand a chance to be accepted. However, in its current form, I feel this work is not ready for publication.

==== Update ====

The rebuttal addressed my main concern. For that reason, I raised my score from 3 to 5. Depending on how discussion among reviewers and the AC goes, I might further increase my score to 6. The reason I'm not considering giving the paper a higher score is due to the issue I raised regarding the fact that the model might be of somewhat limited interest.

[KV18] Vishesh Karwa and Salil P. Vadhan. Finite sample differentially private confidence intervals. In
ITCS, pages 44:1–44:9, 2018

[RSSS21] Sofya Raskhodnikova, Satchit Sivakumar, Adam Smith, and Marika Swanberg. Differentially
private sampling from distributions. In NeurIPS, pages 28983–28994, 2021.

**Questions:**

I guess the one question I have has to do with the main issue I raised in the section about weaknesses. Is there something that I have missed or misjudged in terms of the dependencies in the range parameters? If I have misunderstood something and the authors can clarify things to me, I am open to raising my score.

**Limitations:**

The work is purely theoretical and doesn't have any obvious prospects for societal impact (positive or negative).

---

> ### Author Rebuttal · Authors · 2023-08-08
>
> We wish to correct the misunderstanding here: Our bounds are completely *independent of the $R$ parameter* and in fact *do not* require any assumption on $R$. Please refer to Theorems 1.2, 1.4, and 1.6 for our bounds.
>
> It seems like the misunderstanding is perhaps due to focusing on the bounds in Section 4.2. However, as we point out in Section 4.1, it is simple to use a known result in literature to reduce the general case to the case where $R$ is bounded. In particular, Lemma 4.1 essentially says that we may assume w.l.o.g. that $R \\leq O(\\kappa \\cdot \\sqrt{d})$. As we stated in the paper, combining Lemma 4.1 with the bounds in Section 4.2 (Theorem 4.2 and Theorem 4.3 respectively) immediately imply the bounds that are *independent* of $R$ (Theorem 1.2 and Theorem 1.4, respectively).  Apologies for causing the confusion; in the revision, we will reword this more clearly.
>
> Due to this misunderstanding, we kindly ask the reviewer to adjust the score accordingly.
>
> Regarding the dependency on $\\kappa$, indeed our Theorem 1.4 for the bounded unknown covariance case has a polynomial dependency on $\\kappa$ but we wish to point out that the other two main theorems (Theorem 1.2, Theorem 1.6) *do not* have any dependency on $\kappa$. As we noted in footnote 6 (page 8), it is possible to reduce the dependency in Theorem 1.4 to polylogarithmic in $\\kappa$ but we have to increase the dependency on $d$ to $d^{1.5}$. It remains an interesting question whether one can get $polylog(\\kappa)$ dependency while having linear dependency on $d$. We will make sure to highlight it as an open question in the revision.

---

> > ### Comment · Reviewer_cbpB · 2023-08-21
> >
> > Thank you for your response! I have raised my score and may raise it a bit more during the discussion period. See the update in my review for more details.

---

### Official Review · Reviewer_nVvL · 2023-07-11

**Soundness:** 4 excellent
**Presentation:** 4 excellent
**Contribution:** 4 excellent
**Rating:** 7
**Confidence:** 4

**Summary:**

Raskhodnikova, Sivakumar, Smith, and Swanberg (NeurIPS '21) recently introduced the problem of privately approximate sampling: Given i.i.d. draws from a member D of a known family of distributions, produce one sample whose distribution is close to D in TV distance. This is an easier task than the well-studied problem of private distribution learning, while isolating some of its algorithmic difficulty, and RSSS'21 gave several upper and lower bounds for discrete distributions and product distributions.

This submission addresses several open directions from RSSS'21 by 1) thoroughly investigating the sample complexity of privately sampling from Gaussians, and 2) giving a pure DP sampler for Bernoulli product distributions with optimal sample complexity (the sample-optimal algorithm from RSSS'21 only guaranteed approximate DP). In more detail, this paper establishes tight (up to polylog factors) sample complexity upper and lower bounds for sampling from Gaussians with either known or a prior bounded covariance, as well as giving an upper bound in the unbounded covariance case that separates DP sampling from DP learning.

**Strengths:**

- The problem of private sampling is reasonably well-motivated from the standpoint of "applications" (i.e., minimizing the overhead of privacy when full distribution learning is not necessary) as well as from the standpoint of doing theory, i.e., developing a finer understanding of what makes private statistical analysis challenging, establishing new algorithmic primitives that can be used elsewhere, and more generally forcing us to develop new techniques.

- Within the context above, this paper addresses two of the most fundamental classes of distributions. In several cases, it identifies nearly matching upper and lower bounds.

- The paper makes the interesting finding that, at least for Gaussian sampling, the dependence on the error parameter $\alpha$ is at most polylogarithmic. In contrast, the dependence is at least polynomial for learning Gaussians, as well as for the discrete sampling tasks studied previously.

- Both the algorithms and lower bounds exhibit interesting (and in my opinion, surprising) technical insights. For instance, Algorithm 2 for learning Gaussians with unknown bounded covariance describes and analyzes a new sampling distribution based on taking linear combinations of spherical noise. As another example, the lower bound for the same setting implements a conceptually very natural reduction, but has to overcome a technical issue of breaking the correlation between samples generated by running a private sampler multiple times on the same dataset (which addresses a more general issue with DP sampling and could be used in other contexts).

**Weaknesses:**

- The algorithms for Gaussian sampling are quite specialized to Gaussians, and it's quite difficult to imagine how they might generalize to other classes of distributions.

- This is sort of a nitpick, but I think the writing could highlight some of the new ideas described under "strengths" above, e.g., with lemma statements that would make them easier to apply elsewhere.

**Questions:**

- On the topic of generalizing to other distributions: Many algorithms for learning distributions (including Gaussians) readily generalize to the semi-agnostic case, incurring additional error that scales with the amount of model misspecification. Does such a ready generalization apply to the algorithms in this paper?

- Lines 81-82 mention that the paper's algorithms run in polynomial time. In what computational model? Since the problems considered require TV-close sampling from continuous distributions, it's not clear that taking a fixed precision discretization suffices.

- Is it possible to incorporate some polylogarithmic in $\alpha$ dependence on the error in the lower bounds for Gaussian sampling? (As things stand, it could be the case that the right dependence on $\alpha$ is, say, doubly logarithmic.)

**Limitations:**

The paper's conclusion raises the question about extending beyond Gaussian distributions mentioned above, as well as the problem of closing the quantitative gap between upper and lower bounds for the unbounded covariance case.

---

> ### Author Rebuttal · Authors · 2023-08-08
>
> We thank the reviewer for their review & comments. Please find our answers to the questions below.
>
> - It is possible to get some bounds in the agnostic case using our algorithms, but these bounds are not particularly good. In particular, let $n$ be the sample complexity in our algorithms (e.g., from Theorems 1.2, 1.4, or 1.6). Then, if we just run the same algorithm for some distribution $D$ that is $\gamma$-close in TV distance to a Gaussian (in the corresponding setting), then the accuracy parameter will increase to $\alpha + n \gamma$.
> - For simplicity, we assume that our algorithm can add, subtract, multiply, and divide real numbers; furthermore, we assume that we can sample from a (one-dimensional) Gaussian distribution exactly. We note that it is a fairly standard assumption in DP literature when continuous (e.g., Laplace / Gaussian) noise is involved. It is possible to make our algorithm work in the finite-precision setting but this is quite complicated (the accuracy guarantee will need to be with respect to the discrete instead of the continuous Gaussians, or in the Kolmogorov distance instead of the total variation distance).
> - This is a very interesting question. We are not aware of any lower bounds involving $\\alpha$ and we will include this as an open question in the revision.

---

> > ### Comment · Reviewer_nVvL · 2023-08-15
> >
> > Thank you for the responses!
> >
> > > For simplicity, we assume that our algorithm can add, subtract, multiply, and divide real numbers; furthermore, we assume that we can sample from a (one-dimensional) Gaussian distribution exactly. We note that it is a fairly standard assumption in DP literature when continuous (e.g., Laplace / Gaussian) noise is involved. It is possible to make our algorithm work in the finite-precision setting but this is quite complicated (the accuracy guarantee will need to be with respect to the discrete instead of the continuous Gaussians, or in the Kolmogorov distance instead of the total variation distance).
> >
> > Thanks, this makes sense. I'd encourage the authors to include a sentence mentioning that they're working in a real-arithmetic model of computation throughout, especially since _a priori_, one might at least expect the algorithm for learning binary product distributions to be implementable on a finite computer. For many DP algorithms solving discrete problems, it's "challenging, but doable" to adapt them to finite precision (see, e.g., papers of Mironov; Balcer and Vadhan; Ilvento; Canonne, Kamath, and Steinke), so at least for theoretical work, the distinction isn't so important; but for tasks like learning Gaussians for which this isn't necessarily the case, my two cents are that it's worth alerting the reader to what they're getting, even if a real model of computation is standard in this subspace of the literature.

---

> > > ### Author Response · Authors · 2023-08-19
> > >
> > > Thank you for your reply. We will incorporate the discussion regarding the model into the revision.

---

### Official Review · Reviewer_jqWC · 2023-07-12

**Soundness:** 4 excellent
**Presentation:** 3 good
**Contribution:** 4 excellent
**Rating:** 7
**Confidence:** 3

**Summary:**

This paper considers a very interesting problem: Design a $\mathit{differentially~private}$ algorithm that given samples from unknown distribution $D$ from a class of distributions ${\mathcal D}$, output a sample from a distribution $D'$ where $D'$ is close to $D$ in total variation distance. One direct approach is to first learn the distribution $D$ in a differentially private manner and output a sample from the distribution. However, the paper argues that learning is an overkill. The paper shows that the sample complexity of DP-sampling problem is substantially less than the DP-learning problem in many cases. In particular, they design algorithms for DP-sampling algorithms of $d$-dimensional Gaussian distributions with optimal sample complexities. They consider three subclasses of Gaussian distributions: (1) known covariance (2) unknown but bounded covariance (3) unknown unbounded covariance -- and sample optimal DP-sampling algorithms for these class of distributions. They also consider the class of product distributions and give optimal pure DP-samplers. Earlier, only approximate DP-samplers where known.

**Strengths:**

The problem they consider is very interesting to me: producing samples from unknown distribution in a differentially private manner appears to be a very basic task. Investigating this problem for Gaussian distributions should be of broad interest.

**Weaknesses:**

I am not sure whether it is a weakness, all proofs are in the appendix and are for specialist so may not be accessible to general audience.

**Questions:**

I do not have any major questions at this point.

**Limitations:**

This is a theoretical paper and does not seem to have any negative societal impact.

---

> ### Author Rebuttal · Authors · 2023-08-08
>
> We thank the reviewer for their review & comments.

---

### Decision · Program_Chairs · 2023-09-21

**Decision:**

Accept (poster)

**Comment:**

The authors consider the problem of differentially private sampling from an unknown distribution (which is easier than learning that distribution). The authors prove multiple results about private sampling from Gaussians and product distributions.

The reviewers found the results interesting and impactful, and in some cases surprising (e.g., the polylogarithmic dependence of the sample complexity of private Gaussian sampling on the accuracy parameter).

Based on these, I recommend acceptance.